# NAFS: A Simple yet Tough-to-beat Baseline for Graph Representation Learning

## Abstract

Recently, graph neural networks (GNNs) have shown prominent performance in graph representation learning by leveraging knowledge from both graph structure and node features. However, most of them have two major limitations. First, GNNs can learn higher-order structural information by stacking more layers but can not deal with large depth due to the over-smoothing issue. Second, it is not easy to apply these methods on large graphs due to the expensive computation cost and high memory usage. In this paper, we present node-adaptive feature smoothing (NAFS), a simple non-parametric method that constructs node representations *without parameter learning*. NAFS first extracts the features of each node with its neighbors of different hops by *feature smoothing*, and then adaptively combines the smoothed features. Besides, the constructed node representation can further be enhanced by the ensemble of smoothed features extracted via different smoothing strategies. We conduct experiments on four benchmark datasets on two different application scenarios: node clustering and link prediction. Remarkably, NAFS with feature ensemble outperforms the state-of-the-art GNNs on these tasks and mitigates the aforementioned two limitations of most learning-based GNN counterparts.

## 1 Introduction

In recent years, graph representation learning has been extensively applied in various application scenarios, such as node clustering, link prediction, node classification, and graph classification (Kipf & Welling, 2016b;a; Hamilton et al., 2017; Bo et al., 2020; Hettige et al., 2020; Wang et al., 2016; Wu et al., 2020; Abu-El-Haija et al., 2019). The goal of graph representation learning is to encode graph information to node embeddings. Traditional graph representation learning methods, such as DeepWalk (Perozzi et al., 2014), Node2vec (Grover & Leskovec, 2016), LINE (Tang et al., 2015), and ComE (Cavallari et al., 2017) merely focus on preserving graph structure information. GNN-based graph representation learning has attracted intensive interest by combining knowledge from both graph structure and node features. While most of these GNN-based methods are designed based on Graph AutoEncoder (GAE) and Variational Graph AutoEncoder (VGAE) (Kipf & Welling, 2016b), these methods share two major limitations:

**Shallow Architecture.** Previous work shows that although stacking multiple GNN layers in Graph Convolutional Network (GCN) (Kipf & Welling, 2016a) is capable of exploiting deep structural information, applying a large number of GNN layers might lead to indistinguishable node embeddings, i.e., the over-smoothing issue (Li et al., 2018). Therefore, most state-of-the-art GNNs resort to shallow architectures, which hinders the model from capturing long-range dependencies.

**Low Scalability.** GNN-based graph representation learning methods can not scale well to large graphs due to the expensive computation cost and high memory usage. Most existing GNNs need to repeatedly perform the computationally expensive and recursive feature smoothing, which involves the participation of the entire graph at each training epoch. Furthermore, most methods adopt the same training loss function as GAE, which introduces high memory usage by storing the dense-form adjacency matrix on GPU. For a graph of size 200 million, its dense-form adjacency matrix requires a space of roughly 150GB, exceeding the memory capacity of the current powerful GPU devices.

To tackle these issues, we propose a new graph representation learning method, which is embarrassingly simple: just smooth the node features and then combine the smoothed features in a node-adaptive manner. We name this method node-adaptive feature smoothing (NAFS), and its

goal is to construct better node embeddings that integrate the information from both graph structural information and node features. Based on the observation that different nodes have highly diverse "smoothing speed", NAFS adaptively smooths each node feature and takes advantage of both low-order and high-order neighborhood information of each node. In addition, feature ensemble is also employed to combine the smoothed features extracted via different smoothing operators. Since NAFS is training-free, it significantly reduces the training cost and scales better to large graphs than most GNN-based graph representation learning methods.

This paper is not meant to diminish the current advancements in GNN-based graph representation learning approaches. Instead, we aim to introduce an easier way to obtain high-quality node embeddings and understand the source of performance gains of these approaches better. Feature smoothing could be a promising direction towards a more simple and effective integration of information from both graph structure and node features.

Our contributions are as follows: (1) *New perspective*. To the best of our knowledge, we are the first to explore the possibility that simple feature smoothing without any trainable parameters could even outperform state-of-the-art GNNs; this incredible finding opens up a new direction towards efficient and scalable graph representation learning. (2) *Novel method*. We propose NAFS, a node-adaptive feature smoothing approach along with various feature ensemble strategies, to fully exploit knowledge from both the graph structure and node features. (3) *State-of-the-art performance*. We evaluate the *effectiveness* and *efficiency* of NAFS on real-world datasets across various graph-based tasks, including node clustering and link prediction. Empirical results demonstrate that NAFS performs comparably with or even outperforms the state-of-the-art GNNs, and achieves up to two orders of magnitude speedup. In particular, on PubMed, NAFS outperforms GAE (Kipf & Welling, 2016b) and AGE (Cui et al., 2020) by a margin of $9.0\%$ and $3.8\%$ in terms of NMI in node clustering, while achieving up to $65.4\times$ and $88.6\times$ training speedups, respectively.

## 2 PRELIMINARY

In this section, we first explain the notations and problem formulation. Then, we review current GNNs and GNN-based graph representation learning.

### 2.1 NOTATIONS AND PROBLEM FORMULATION.

In this paper, we consider an undirected graph $\mathcal{G} = (\mathcal{V}, \mathcal{E})$ with $|\mathcal{V}| = n$ nodes and $|\mathcal{E}| = m$ edges. Here we suppose that $m \propto n$ as it is the case in most real-world graphs. We denote by $\mathbf{A}$ the adjacency matrix of $\mathcal{G}$. Each node can possibly have a feature vector of size $f$, which stacks up to an $n \times f$ feature matrix $\mathbf{X}$. The degree matrix of $\mathbf{A}$ is denoted as $\mathbf{D} = \mathrm{diag}\,(d_1, d_2, \cdots, d_n) \in \mathbb{R}^{n \times n}$, where $d_i = \sum_{v_j \in \mathcal{V}} \mathbf{A}_{ij}$. We denote the final node embedding matrix as $\mathbf{Z}$, and evaluate it in both the node clustering and the link prediction tasks. The node clustering task requires the model to partition the nodes into $c$ disjoint groups $G_1, G_2, \cdots, G_c$, where similar nodes should be in the same group. The target of the link prediction task is to predict whether an edge exists between given node pairs.

### 2.2 GRAPH CONVOLUTIONAL NETWORK.

Based on the assumption that locally connected nodes are likely to enjoy high similarity (McPherson et al., 2001), each node in most GNN models iteratively smooths the representations of its neighbors for better node embedding. Below is the formula of the $l$-th GCN layer (Kipf & Welling, 2016a):

$$\mathbf{X}^{(l)} = \delta\big(\hat{\mathbf{A}}\mathbf{X}^{(l-1)}\boldsymbol{\Theta}^{(l)}\big), \quad \hat{\mathbf{A}} = \widetilde{\mathbf{D}}^{-1/2}\widetilde{\mathbf{A}}\widetilde{\mathbf{D}}^{-1/2}, \quad \widetilde{\mathbf{A}} = \mathbf{A} + \mathbf{I}_n, \tag{1}$$

where $\mathbf{X}^{(l)}$ is the node embedding matrix at layer $l$, $\mathbf{X}^{(0)}$ is the original feature matrix, $\boldsymbol{\Theta}^{(l)}$ are the trainable weights, and $\delta$ is the activation function. $\hat{\mathbf{A}}$ is the smoothing matrix that helps each node to smooth representations of neighboring nodes.

As shown in Eq. 1, each GCN layer contains two operations: feature aggregation (smoothing) and feature transformation. Figure 1 shows the framework of a two-layer GCN. The $l$-th layer in GCN firstly executes feature smoothing on the node embedding $\mathbf{X}^{(l-1)}$. Then, the smoothed feature $\widetilde{\mathbf{X}}^{(l-1)}$ is transformed with trainable weights $\boldsymbol{\Theta}^{(l)}$ and activation function $\delta$ to generate new node embedding $\mathbf{X}^{(l)}$. Note that GCN will degrade to MLP if feature smoothing is removed from each layer.

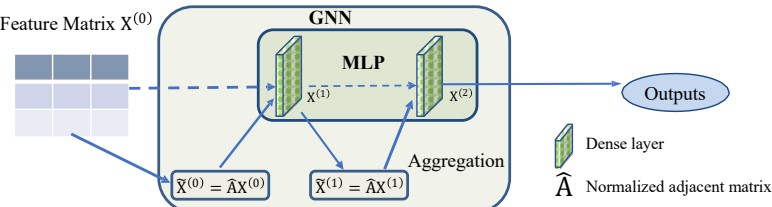

Figure 1: The framework of a two-layer GNN models.

## 2.3 GNN-BASED GRAPH REPRESENTATION LEARNING.

GAE (Kipf & Welling, 2016b), the first and the most representative GNN-based graph embedding method, adopts an encoder to generate node embedding matrix $\mathbf{Z}$ with inputs $\hat{\mathbf{A}}$ and $\mathbf{X}$. A simple inner product decoder is then used to reconstruct the adjacency matrix. The final training loss of GAE is the binary entropy loss between $\mathbf{A}'$ and $\widetilde{\mathbf{A}}$, the reconstructed adjacency matrix and the original adjacency matrix with self loop added.

$$\mathbf{A}' = \text{sigmoid}(\mathbf{Z} \cdot \mathbf{Z}^\mathsf{T}), \quad \mathcal{L} = \sum_{1 \le i,j \le n} -\widetilde{\mathbf{A}}_{i,j} \log \mathbf{A}'_{i,j} - (1 - \widetilde{\mathbf{A}}_{i,j}) \log(1 - \mathbf{A}'_{i,j})). \tag{2}$$

Motivated by GAE, lots of GNN-based graph representation learning methods are proposed recently. MGAE (Wang et al., 2017) presents a denoising marginalized autoencoder that reconstructs the node feature matrix $\mathbf{X}$. ARGA (Pan et al., 2018) adopts the adversarial learning strategy, and its generated node embeddings are forced to match a prior distribution. DAEGC (Wang et al., 2019) exploits side information to generate node embeddings in a self-supervised way. AGC (Zhang et al., 2019) proposes an improved filter matrix to better filter out the high-frequency noise. AGE (Cui et al., 2020) further improves AGC by using the similarity of embedding rather than the adjacency matrix to consider original node feature information. Compared with GNN-based graph representation learning methods that rely on the trainable parameters to learn node embeddings, our NAFS is training-free and thus enjoys higher efficiency and scalability.

## 3 OBSERVATION AND INSIGHT

In this section, we make a quantitative analysis on the over-smoothing issue at the node level and then provide some insights when designing NAFS on graphs.

### 3.1 FEATURE SMOOTHING IN DECOUPLED GNNS

Recently, many works (Wu et al., 2019; Zhu & Koniusz, 2021; Chen et al., 2020; Zhang et al., 2021) propose to decouple the feature smoothing and feature transformation in each GCN layer for scalable node classification. Concretely, they execute the feature smoothing operation in advance, and the smoothed features are then fed into a simple MLP to generate the final predicted node labels. Under this framework, the predictive node classification accuracy of these methods is comparable with or even higher than the one of coupled GNNs, and these works claim that the true power of GNNs lies in feature smoothing rather than feature transformation.

We split the framework of these decoupled GNNs into two parts: feature smoothing and MLP training. Feature smoothing aims to combine the graph structural information and node features into better features for the subsequent MLP; while MLP training only takes in the smoothed feature and is specially trained for a given task. As stated by previous decoupled GNNs (Wu et al., 2019; Zhu & Koniusz, 2021), the true success of GNNs lies in feature smoothing rather than feature transformation. Correspondingly, we propose to remove feature transformation and preserve the key feature smoothing part alone for simple and scalable node representation.

There is another branch of GNNs that also decouple the feature smoothing and feature transformation. The most representative method of this category is APPNP (Klicpera et al., 2018). It first feeds the raw node features into an MLP to generate intermediate node embeddings; then the personalized PageRank based propagation operations are performed on the node embeddings to produce final

prediction results. However, compared with scalable decoupled GNNs mentioned in the previous paragraph, this branch of GNNs still have to recursively execute propagation operations in each training epoch, which makes it impossible to perform on large-scale graphs. In the remaining part of this paper, the terminology "decoupled GNNs" refers particularly to the scalable decoupled GNNs mentioned in the previous two paragraphs.

## 3.2 MEASURING SMOOTHING LEVEL

To capture deep graph structural information, a straightforward way is to simply stack multiple GNN layers. However, a large number of feature smoothing operations in a GNN model would lead to indistinguishable node embeddings, i.e., the over-smoothing issue (Li et al., 2018). Concretely, if we execute $\hat{\mathbf{A}}\mathbf{X}$ for infinite times, the node embeddings within the same connected component would reach a stationary state. When adopting $\hat{\mathbf{A}} = \widetilde{\mathbf{D}}^{r-1}\widetilde{\mathbf{A}}\widetilde{\mathbf{D}}^{-r}$, $\hat{\mathbf{A}}^{\infty}$ follows

$$\hat{\mathbf{A}}_{i,j}^{\infty} = \frac{(d_i + 1)^r (d_j + 1)^{1-r}}{2m + n}, \tag{3}$$

which shows that the influence from node $v_i$ to $v_j$ is only determined by their degrees. Under the extreme condition that $r = 0$, all the nodes within one connected component have exactly the same representation, making it impossible to apply the node embeddings to subsequent tasks.

Here we introduce a new metric, "Over-smoothing Distance", to measure each node's smoothing level. A smaller value indicates that the node is closer to the stationary state, i.e., closer to over-smoothing.

**Definition 3.1** (**Over-smoothing Distance**). *The Over-smoothing Distance $D_i(k)$ parameterized by node $i$ and smoothing step $k$ is defined as*

$$D_i(k) = Dis([\hat{\mathbf{A}}^k\mathbf{X}]_i, [\hat{\mathbf{A}}^{\infty}\mathbf{X}]_i), \tag{4}$$

*where $[\hat{\mathbf{A}}^k\mathbf{X}]_i$ denotes the $i^{th}$ row of $\hat{\mathbf{A}}^k\mathbf{X}$, representing the representations of node $v_i$ after smoothing $k$ times; $[\hat{\mathbf{A}}^{\infty}\mathbf{X}]_i$ denotes the $i^{th}$ row of $\hat{\mathbf{A}}^{\infty}\mathbf{X}$, representing the stationary state of node $v_i$; $Dis(\cdot)$ is a distance function or a function positively relative with the difference, which can be implemented using Euclidean distance, the inverse of cosine similarity, etc.*

## 3.3 DIVERSE SMOOTHING SPEED ACROSS NODES.

To examine the factors that affect $D_i(k)$, we divide nodes in the PubMed dataset into three different groups according to their degrees. In Figure 2, we show the trends of nodes in the first group and the second group where the group-averaged $D_i(k)$ changes with the number of smoothing step increases. The trend of $D_i(k)$ averaged over all the nodes is also provided, and the Euclidean distance is chosen as the distance function. Figure 2 shows that the $D_i(k)$ of nodes with degrees larger than 60 drops more quickly than nodes with degrees smaller than 3, which implies that nodes with high degrees approach the stationary state more rapidly than the nodes with low degrees.

The quantitative analysis empirically illustrates that the degree of each node plays an essential role in one's optimal smoothing step. Intuitively, nodes with high degrees should have relatively small smoothing steps than nodes with low degrees. In addition, in Appendix A, we have made a detailed theoretical analysis about the graph sparsity, another factor that influences the smoothing speed.

## 3.4 DESIGN INSIGHTS OF NAFS

Though using feature smoothing operations inside the previous decoupled GNNs is scalable for large graph representation learning, it will lead to sub-optimal node representation. According to the observation in Sec. 3.3, it is sub-optimal to execute feature smoothing for all the nodes indiscriminately as previous decoupled GNNs do since nodes with different structural properties have diverse smoothing speeds. Therefore, node-adaptive feature smoothing (i.e., different nodes have different smoothing levels or mechanisms with equivalent effect) must be adopted to satisfy each node's diverse needs of smoothing level. In our proposed method NAFS, we utilize the metric $D_i(k)$ defined in Def. 3.1 to help accomplish the aim of node-adaptive feature smoothing.

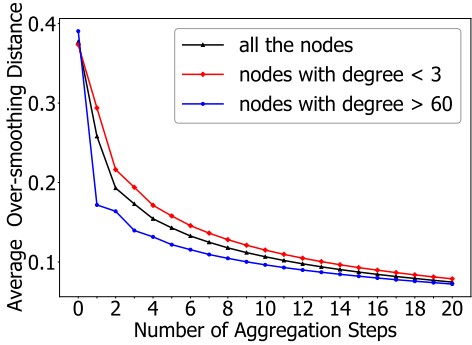

Figure 2: Diverse smoothing speed across nodes with different degrees.

## 4 PROPOSED METHOD

In this section, we present NAFS, a training-free method for scalable graph representation learning. We first compute the smoothed features with the feature smoothing operation. Then the feature ensemble operation is used to combine the smoothed features generated by different smoothing strategies. The pseudo code of NAFS is provided in Appendix D.

### 4.1 NODE-ADAPTIVE FEATURE SMOOTHING

Figure 3 provides an overview of NAFS. When repeatedly executing $\mathbf{X}^{(l)} = \hat{\mathbf{A}}\mathbf{X}^{(l-1)}$, the smoothed node embedding matrix $\mathbf{X}^{(l-1)}$ contains deeper graph structural information with $l$ increases. The multi-scale node embedding matrices $\{\mathbf{X}^{(0)}, \mathbf{X}^{(1)}, ..., \mathbf{X}^{(K)}\}$ ($K$ is the maximal smoothing step) are then combined into a single matrix $\hat{\mathbf{X}}$ such that both local and global neighborhood information are reserved for each node.

The analysis in Sec. 3.3 illustrates that the speed each node achieving its stationary state is highly diverse, which suggests that nodes should be treated individually. To this end, we define "Smoothing Weight" based on $D_i(k)$ introduced in Def. 3.1 for each node so that the smoothing operation can be performed in a node-adaptive manner.

**Definition 4.1** (**Smoothing Weight**). *The Smoothing Weight $w_i(k)$ parameterized by node $v_i$ and smoothing step $k$ is defined based on the softmax value of $\{D_i(0), D_i(1), \cdots, D_i(K)\}$:*

$$w_i(k) = e^{D_i(k)} / \sum_{l=0}^{K} e^{D_i(l)}, \tag{5}$$

*where $K$ is the maximal smoothing step.*

To calculate $D_i(k)$ more efficiently, an alternative is to replace $[\hat{\mathbf{A}}^\infty \mathbf{X}]_i$ in Eq. 4 with $\mathbf{X}_i$ and implement $Dis(\cdot)$ as the cosine similarity. Larger $D_i(k)$ in this case means that node $v_i$ is farther from the stationary state and $[\hat{\mathbf{A}}^k \mathbf{X}]_i$ intuitively contains more relevant node information. Therefore, for node $v_i$, the smoothed feature with larger $D_i(k)$ (i.e., larger $w_i(k)$) should contribute more to the final node embedding. The smoothing weight can be formulated in the following matrix form:

**Definition 4.2** (**Smoothing Weight Matrix**). *The Smoothing Weight Matrix $\mathbf{W}(k)$ parameterized by smoothing step $k$ is defined as the diagonal matrix derived from $\eta(k) \in \mathbb{R}^n$:*

$$\mathbf{W}(k) = Diag(\eta(k)), \quad \eta(k)[i] = w_i(k), \quad \forall 1 \le i \le n. \tag{6}$$

Given the maximal smoothing step $K$, the multi-scale smoothed features $\{\mathbf{X}^{(0)}, \mathbf{X}^{(1)}, ..., \mathbf{X}^{(K)}\}$, and the corresponding Smoothing Weight Matrices $\{\mathbf{W}(0), \mathbf{W}(1), ..., \mathbf{W}(K)\}$ , the final smoothed feature $\hat{\mathbf{X}}$ can be represented as: $\hat{\mathbf{X}} = \sum_{k=0}^{K} \mathbf{W}(k)\hat{\mathbf{A}}^k \mathbf{X}$.

### 4.2 FEATURE ENSEMBLE

Different smoothing operators actually act as different knowledge extractors. For example, by setting $r = 0.5, 1$ and $0$, $\hat{\mathbf{A}} = \widetilde{\mathbf{D}}^{r-1}\widetilde{\mathbf{A}}\widetilde{\mathbf{D}}^{-r}$ represents the symmetric normalized adjacency matrix

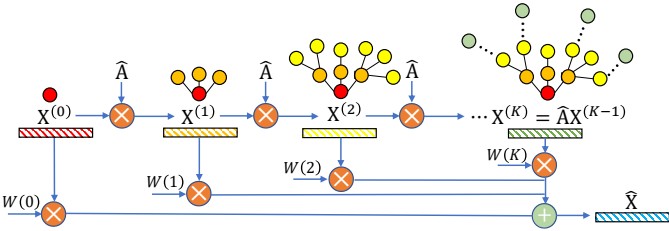

Figure 3: An overview of the proposed node-adaptive feature smoothing operation.

$\widetilde{\mathbf{D}}^{-1/2}\widetilde{\mathbf{A}}\widetilde{\mathbf{D}}^{-1/2}$ (Kipf & Welling, 2016a), the random walk transition probability matrix $\widetilde{\mathbf{D}}^{-1}\widetilde{\mathbf{A}}$ (Xu et al., 2018), and the reverse random walk transition probability matrix $\widetilde{\mathbf{A}}\widetilde{\mathbf{D}}^{-1}$ (Zeng et al., 2019), respectively. These variants of $\hat{\mathbf{A}}$ captures and reserves different scales and dimensions of knowledge from both graph structures and node features.

To achieve the same effect, the feature ensemble operation has multiple knowledge extractors: we vary the value of $r$ in the normalized adjacency matrix $\hat{\mathbf{A}}_r = \widetilde{\mathbf{D}}^{r-1}\widetilde{\mathbf{A}}\widetilde{\mathbf{D}}^{-r}$ to easily acquire different knowledge extractors. These knowledge extractors are adopted inside the feature smoothing operation to generate different smoothed features. Concretely, the value of $r$ controls the normalized weight of each edge. So different values of $r$ generate different weight values for all the edges in the graph, which would increase the diversity of our smoothed features evidently. The ablation study in Sec. 6.5 shows that the increased diversity contributes a lot to the high performance of our generated node embeddings when applied to downstream tasks.

The detailed procedure of feature ensemble operation is as follows: Given $\{r_1, r_2, ..., r_T\}$, we firstly perform the feature smoothing operation to generate corresponding smoothed features $\{\hat{\mathbf{X}}^{(1)}, \hat{\mathbf{X}}^{(2)}, ..., \hat{\mathbf{X}}^{(T)}\}$. Then, we combine them as $\mathbf{Z} \leftarrow \oplus_{i \in \{1,2,...,T\}} \hat{\mathbf{X}}^{(i)}$, where $\oplus$ is the ensemble strategy, which can be implemented as concatenating, mean pooling, max pooling, etc.

## 5 ADVANTAGES OVER TRADITIONAL APPROACHES

NAFS generates node embeddings in a training-free manner, making it highly efficient and scalable. Moreover, the node-adaptive smoothing strategy enables it to capture deep structural information. In this subsection, we analyze the advantages of our NAFS over GAE and its variants.

**Deep Structural Information.** By assigning each node with personalized smoothing weights, NAFS can gather deep structural information without encountering the over-smoothing issue and keep the time and memory cost low. While for GAE and its variants, they either 1) have a coupled structure that cannot go deep due to low efficiency and high memory cost (e.g., GAE) or 2) are unable to adaptively capture structural information and encounter the over-smoothing issue when going deeper (e.g., AGE).

**Efficiency.** Compared with GAE and its variants, our proposed NAFS does not have any trainable parameters, giving it a significant advantage in efficiency. When generating node embeddings, NAFS only needs to execute the feature smoothing and feature ensemble operations, which has a time complexity of $\mathcal{O}(Kmf + Kn)$, where $K$ denotes the maximal smoothing step. On the other hand, the asymptotic training time complexity of GAE and its variants is $\mathcal{O}(nf^2 + n^2f)$, which is one magnitude larger than NAFS. In Sec. 6.4, we further present efficiency comparison in detail on real-world datasets between NAFS and GAE.

**Memory Cost.** Our proposed NAFS enjoys not only high efficiency but also low memory cost. NAFS only requires to store the sparse adjacency matrix and the smoothed features, and thus the memory cost is $\mathcal{O}(m + nf)$, which grows linearly with graph size $n$ in typical real-world graphs. For GAE and its variants, they need additional space to store the training parameters and especially the reconstructed dense adjacency matrix, making their memory cost $\mathcal{O}(n^2 + m + nf)$, which is also one magnitude larger than NAFS. Scalability comparison conducted on synthetic datasets between NAFS and GAE can be found in Appendix C.2.

Table 1: Link prediction performance comparison.

| Methods | Cora | | Citeseer | | PubMed | |
|---|---|---|---|---|---|---|
| | AUC | AP | AUC | AP | AUC | AP |
| SC | 84.6±0.0 | 88.5±0.0 | 80.5±0.0 | 85.0±0.0 | 84.2±0.0 | 87.8±0.0 |
| DeepWalk | 83.1±0.3 | 85.0±0.4 | 80.5±0.5 | 83.6±0.4 | 84.4±0.4 | 84.1±0.5 |
| GAE | 91.0±0.5 | 92.0±0.4 | 89.5±0.3 | 89.9±0.4 | 96.4±0.4 | 96.5±0.5 |
| VGAE | 91.4±0.5 | 92.6±0.4 | 90.8±0.4 | 92.0±0.3 | 94.4±0.5 | 94.7±0.4 |
| ARGA | 92.4±0.4 | 93.2±0.3 | 91.9±0.5 | 93.0±0.4 | 96.8±0.3 | 97.1±0.5 |
| ARVGA | 92.4±0.4 | 92.6±0.4 | 92.4±0.5 | 93.0±0.3 | 96.5±0.5 | 96.8±0.4 |
| GALA | 92.1±0.3 | 92.2±0.4 | 94.4±0.5 | 94.8±0.5 | 93.5±0.4 | 94.5±0.4 |
| AGE | **95.1±0.5** | **94.6±0.5** | **96.3±0.4** | **96.6±0.4** | 94.3±0.3 | 93.5±0.5 |
| NAFS-simple | 91.9±0.0 | 93.1±0.0 | 94.1±0.0 | 95.2±0.0 | 97.4±0.0 | **97.2±0.0** |
| NAFS-mean | 92.6±0.0 | 93.9±0.0 | 94.9±0.0 | 95.9±0.0 | 97.4±0.0 | **97.2±0.0** |
| NAFS-max | 93.0±0.0 | 94.2±0.0 | 94.8±0.0 | 96.0±0.0 | 97.5±0.0 | 97.1±0.0 |
| NAFS-concat | 92.6±0.0 | 93.8±0.0 | 93.7±0.0 | 93.1±0.0 | **97.6±0.0** | **97.2±0.0** |

## 6 EXPERIMENTAL RESULTS

In this section, we conduct extensive experiments to evaluate the proposed NAFS. We first introduce the considered baseline methods, used datasets, and experimental settings. Then, we demonstrate the advantages of NAFS from the following three perspectives: (1) end-to-end comparison with the state-of-the-art methods, (2) scalability and efficiency, and (3) effectiveness along with ablation studies.

### 6.1 DATASETS AND BASELINE METHODS

Four widely-used network datasets (i.e., Cora, Citeseer, PubMed, and Wiki) are used in our experiments. In Cora and Citeseer, the node features are binary word vectors; while in PubMed and Wiki, each node has a TF-IDF weighted word vector. We include the properties of these datasets in Appendix B.1.

For different downstream tasks, the considered baselines are as follows:

- **Node clustering:** GAE and VGAE (Kipf & Welling, 2016b), MGAE (Wang et al., 2017), ARGA and ARVGA (Pan et al., 2018), AGC (Zhang et al., 2019), DAEGC (Wang et al., 2019), and AGE (Cui et al., 2020).
- **Link prediction:** Spectral Clustering (SC) (Ng et al., 2002), DeepWalk (Perozzi et al., 2014), GAE and VGAE (Kipf & Welling, 2016b), ARGA and ARVGA (Pan et al., 2018), GALA (Park et al., 2019), and AGE (Cui et al., 2020).

For NAFS, we investigate four variants: NAFS-simple, NAFS-mean, NAFS-max, and NAFS-concat. NAFS-simple only contains the feature smoothing operation. Besides, it only uses the smoothed features at maximal smoothing step $K$ with $r = 0.5$ and excludes the node-adaptive strategy. And for NAFS-mean, -max, and -concat, they all have the complete NAFS framework and only differ in the ensemble strategy adopted in feature ensemble operation.

### 6.2 EXPERIMENTAL SETTINGS

**Node Clustering.** For the node clustering task, we apply K-Means (Hartigan & Wong, 1979) to node embeddings to get the clustering results. Three widely-used metrics are used for evaluation: Accuracy (ACC), Normalized Mutual Information (NMI), and Adjusted Rand Index (ARI).

**Link Prediction.** For the link prediction task, 5% and 10% edges are randomly reserved for the validation set and the test set. Once the node embeddings have been generated, we follow the same procedure as GAE to reconstruct the adjacency matrix. Two metrics - Area Under Curve (AUC) and Average Precision (AP) are used in the evaluation of the link prediction task.

**Hyperparameters.** When generating node embeddings, we use the values of $r$ in [0, 0.1, 0.2, 0.3, 0.4, 0.5] to get six different normalized adjacency matrix $\hat{\mathbf{A}}$. The only exception is the link prediction task on the PubMed dataset, where we use values of $r$ in [0.3, 0.4, 0.5] instead. The optimal value of

Table 2: Node clustering performance comparison.

| Methods | Cora | | | Citeseer | | | PubMed | | | Wiki | | |
|---|---|---|---|---|---|---|---|---|---|---|---|---|
| | ACC | NMI | ARI | ACC | NMI | ARI | ACC | NMI | ARI | ACC | NMI | ARI |
| GAE | 53.3±0.2 | 40.7±0.3 | 30.5±0.2 | 41.3±0.4 | 18.3±0.3 | 19.1±0.3 | 63.1±0.4 | 24.9±0.3 | 21.7±0.2 | 37.9±0.2 | 34.5±0.3 | 18.9±0.2 |
| VGAE | 56.0±0.3 | 38.5±0.4 | 34.7±0.3 | 44.4±0.2 | 22.7±0.3 | 20.6±0.3 | 65.5±0.2 | 25.0±0.4 | 20.3±0.2 | 45.1±0.4 | 46.8±0.3 | 26.3±0.4 |
| MGAE | 63.4±0.5 | 45.6±0.3 | 43.6±0.4 | 63.5±0.4 | 39.7±0.4 | 42.5±0.5 | 59.3±0.5 | 28.2±0.2 | 24.8±0.4 | 52.9±0.3 | 51.0±0.4 | 37.9±0.5 |
| ARGA | 63.9±0.4 | 45.1±0.3 | 35.1±0.5 | 57.3±0.5 | 35.2±0.3 | 34.0±0.4 | 68.0±0.5 | 27.6±0.4 | 29.0±0.4 | 38.1±0.5 | 34.5±0.3 | 11.2±0.4 |
| ARVGA | 64.0±0.5 | 44.9±0.4 | 37.4±0.5 | 54.4±0.5 | 25.9±0.5 | 24.5±0.3 | 51.3±0.4 | 11.7±0.3 | 7.8±0.2 | 38.7±0.4 | 33.9±0.4 | 10.7±0.2 |
| AGC | 68.9±0.5 | 53.7±0.3 | 48.6±0.3 | 66.9±0.5 | 41.1±0.4 | 41.9±0.5 | 69.8±0.4 | 31.6±0.3 | 31.8±0.4 | 47.7±0.3 | 45.3±0.5 | 34.3±0.4 |
| DAEGC | 70.2±0.4 | 52.6±0.3 | 49.7±0.4 | 67.2±0.5 | 39.7±0.5 | 41.1±0.4 | 66.8±0.5 | 26.6±0.2 | 27.7±0.3 | 48.2±0.4 | 44.8±0.4 | 33.1±0.3 |
| AGE | 72.8±0.5 | 58.1±0.6 | 56.3±0.4 | 70.0±0.3 | 44.6±0.4 | 45.4±0.5 | 69.9±0.5 | 30.1±0.4 | 31.4±0.6 | 51.1±0.6 | 53.9±0.4 | 36.4±0.5 |
| NAFS-simple | 66.4±0.0 | 50.8±0.0 | 41.8±0.0 | 71.5±0.0 | 44.4±0.0 | 46.8±0.0 | 63.0±0.0 | 27.9±0.0 | 25.1±0.0 | 44.8±0.0 | 43.1±0.0 | 20.0±0.0 |
| NAFS-mean | 70.4±0.0 | 56.6±0.0 | 48.0±0.0 | 71.8±0.0 | 45.1±0.0 | 47.6±0.0 | 70.5±0.0 | 33.9±0.0 | 33.2±0.0 | 54.6±0.0 | 49.4±0.0 | 27.3±0.0 |
| NAFS-max | 70.8±0.0 | 56.6±0.0 | 49.0±0.0 | 70.1±0.0 | 45.1±0.0 | 44.7±0.0 | 70.6±0.0 | 33.4±0.0 | 33.1±0.0 | 51.4±0.0 | 45.8±0.0 | 25.5±0.0 |
| NAFS-concat | 75.4±0.0 | 58.6±0.0 | 53.8±0.0 | 71.1±0.0 | 45.8±0.0 | 46.1±0.0 | 70.5±0.0 | 33.9±0.0 | 33.2±0.0 | 53.6±0.0 | 50.5±0.0 | 26.3±0.0 |

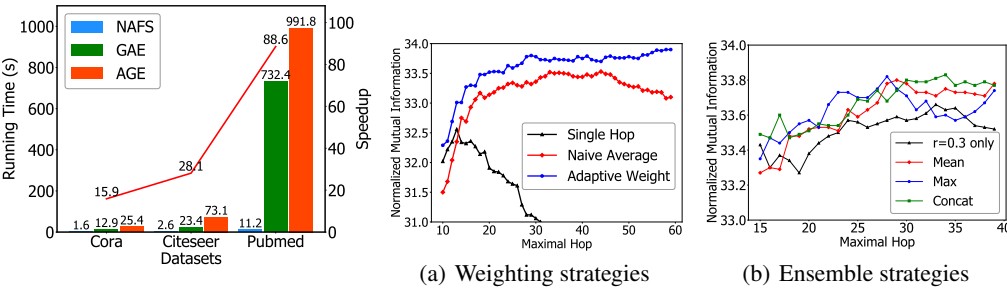

Figure 4: Efficiency comparison on citation networks.

(a) Weighting strategies  (b) Ensemble strategies

Figure 5: Ablation study.

the maximal smoothing steps ranges from 1 to 70. Hyperparameters for all the baseline methods are tuned following the settings in their original paper. We run all the methods using 200 epochs and repeat 10 times on all the datasets, and report the mean value of each evaluation metric. The details of the experimental environment setting are in Appendix B.2.

### 6.3 END-TO-END COMPARISON

**Link Prediction.** Table 1 shows the performance of different methods on the link prediction task. On the three datasets, the proposed NAFS consistently achieves the best or the second-best performance compared with all the baseline methods. Remarkably, although NAFS-simple falls behind the other NAFS variants on Cora and Citeseer, it achieves state-of-the-art performance on PubMed, and outperforms the current state-of-the-art method - ARGA by 0.6% and 0.1% on AUC and AP, respectively.

**Node Clustering.** The node clustering results of each method are shown in Table 2. Among three ensemble strategies, NAFS-concat has the overall best performance across the three datasets, which also consistently outperforms the strongest baseline - AGE. For example, the best of NAFS variants exceeds AGE by 2.6%, 1.8%, 0.7%, and 3.5% on Cora, Citeseer, PubMed, and Wiki, respectively. Although NAFS-concat has the overall best performance, it falls behind NAFS-mean or NAFS-max on some metrics of some datasets; and it needs more CPU memory to store the node embeddings and longer time for inference.

It is quite interesting to find that the simplified version of NAFS, NAFS-simple, outperforms training-based GAE on all the four datasets, and it even outperforms the current SOTA method, AGE, on Citeseer. The competitive performance of NAFS-simple further illustrates that it is possible to achieve decent performance on graphs with only feature smoothing without any training parameters.

### 6.4 EFFICIENCY AND SCALABILITY ANALYSIS

To validate the efficiency and scalability of NAFS, we compare it with GAE and AGE, measuring their overall running time for generating node embeddings. The comparison is conducted on the three citation networks. For fairness, all methods only use CPUs for computation. Figure 4 showcases the results along with the speedup ratio of NAFS against AGE.

Results from Figure 4 show that NAFS is significantly faster than the considered methods on the three datasets. For example, NAFS is almost two magnitudes faster than AGE on the relatively large dataset - PubMed. The high efficiency of NAFS is attributed to no trainable parameters, while GAE and AGE are both training-based methods. Although AGE executes the feature smoothing step in advance, it adopts a time-consuming ranking policy to select positive and negative examples. This alteration in AGE brings both superior performance and longer running time compared with GAE. In addition, we also include more details in Appendix C.2 to demonstrate the excellent scalability of NAFS (See Figure 6).

## 6.5 ABLATION STUDY

To thoroughly investigate the proposed NAFS, ablation studies on the node clustering task are designed to analyze the effectiveness of feature smoothing and feature ensemble in NAFS. The experiments are conducted on the PubMed dataset, and Normalized Mutual Information (NMI) is used to measure the performance. To interpret its effectiveness better, we also visualize the node embedding in Appendix C.4.

**Different Weighting Strategies.** In NAFS, we use node-adaptive weight to average the smoothed features of different steps. Here we change the "Adaptive Weight" in our method to "Single Hop" (only the smoothed feature at the last smoothing step is reserved) and "Naive Average" (smoothed features at every smoothing step has equal weight) and evaluate their performance. Figure 5(a) shows the performance of the three different weighting strategies.

From Figure 5(a), the weighting strategy "Single Hop" performs the worst among the three, since it only makes use of the information at the last smoothing step, which could lead to the over-smoothing issue when the maximal smoothing step becomes large. On the other hand, the weighting strategy "Adaptive Weight" shows a better performance than "Naive Average". Besides, when the number of maximal smoothing steps becomes large, the performance of "Naive Average" begins to drop, while "Adaptive Weight" does not. "Naive Average" assigns the same weight across all the different steps of smoothed features, which also leads to over-smoothing when it tries to exploit extremely deep structural information. Instead, by assigning adaptive weights, the "Adaptive Weight" strategy in NAFS could exploit such deep information and avoid the over-smoothing issue, thus improving the quality of the generated node embedding.

**Different Ensemble Strategies.** We use different ensemble strategies to obtain the final node embeddings in our proposed method - NAFS, which include "Mean", "Max", and "Concat". To evaluate the impacts of these different ensemble strategies, we change the maximal smoothing step, $K$, from 15 to 40, and evaluate corresponding node clustering performance. For reference, the performance of NAFS without feature ensemble, "r=0.3 only" is also reported. The experimental results in Figure 5(b) illustrate that at most times, "r=0.3 only" performs worse than others, which shows the effectiveness of the three different ensemble strategies. However, if we limit the comparison within the ensemble strategies, the performance superiority is unclear. In Table 2 and 1 we can also have that different ensemble strategies perform diversely across different datasets and tasks. It indicates that these three ensemble strategies all have necessities in their own way.

## 7 CONCLUSION

This paper presents NAFS, a novel graph representation learning method. Unlike other GNN-based approaches, NAFS focuses on improving the feature smoothing operation in the GNN layer and generate node embeddings in a training-free manner. NAFS proposes Node-Adaptive Feature Smoothing to generate smoothed features with adaptivity to each node's individual properties; it further employs feature ensemble to combine multiple smoothed features from diverse knowledge extractors effectively. Experiments results on typical tasks demonstrate that NAFS performs comparably with or even outperforms the state-of-the-art GNNs, and at the same time enjoys high efficiency and scalability where NAFS shows its absolute superiority.

## 8 REPRODUCIBILITY STATEMENT

The source code of NAFS can be found in Anonymous Github (`https://anonymous.4open.science/r/NAFS`). To ensure reproducibility, we have provided the overview of datasets and baselines in Section 6.1 and Appendix B.1. The settings for each task and baseline can be found in Section 6.2. Our experimental environment is presented in Appendix B.2, and please refer to "README.md" in the Github repository for more reproduction details.

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

# A    THEORETICAL ANALYSIS

The essential kernel of NAFS is the Smoothing Weight, which determines the output results. We now analyze the factors affecting the value of Smoothing Weight. To simplify our analysis, we suppose $r = 0$ in the normalized adjacency matrix and apply Euclidean distance as the distance function in Definition 3.1. Thus we have

$$D_i(k) = ||[\hat{\mathbf{A}}^k \mathbf{X}]_i - [\hat{\mathbf{A}}^\infty \mathbf{X}]_i||_2, \tag{7}$$

where $|| \cdot ||_2$ symbols two-norm.

**Theorem A.1.** *For any node $i$ in graph $\mathcal{G}$, there always exists*

$$D_i(k) \leq \lambda_2^k \sqrt{\frac{\sum\limits_{j=1}^{n} (\tilde{d}_j ||\mathbf{X}_j||_2^2)}{\tilde{d}_i}}, \tag{8}$$

*where $\tilde{d}_i = d_i + 1$, $\tilde{d}_j = d_j + 1$, $||\mathbf{X}_j||_2$ denotes the two-norm of the initial feature of node $j$, and $0 < \lambda_2 < 1$ denotes the second largest eigenvalue of the normalized adjacency matrix $\hat{\mathbf{A}}$.*

To prove Theorem A.1, we introduce the following lemma.

**Lemma A.2.**

$$|(e_i \hat{\mathbf{A}}^k)_j - (e_i \hat{\mathbf{A}}^\infty)_j| \leq \sqrt{\frac{\tilde{d}_j}{\tilde{d}_i}} \lambda_2^k, \tag{9}$$

*where $e_i$ denotes a one-hot row vector with its $i^{th}$ components as 1 and other components as 0, $\lambda_2$ is the second largest eigenvalue of $\hat{\mathbf{A}}$ and $\tilde{d}_i$ denotes the degree of node $i$ plus 1 (to include itself).*

$$\tilde{d}_i = d_i + 1, \quad \tilde{d}_j = d_j + 1,$$

The proof of Lemma A.2 can be found in (Chung & Graham, 1997). Next we will prove Theorem A.1.

*Proof.* According to equation 7, we can have that

$$
\begin{aligned}
D_i(k) &= ||[\hat{\mathbf{A}}^k \mathbf{X}]_i - [\hat{\mathbf{A}}^\infty \mathbf{X}]_i||_2 \\
&= ||(e_i \hat{\mathbf{A}}^k - e_i \hat{\mathbf{A}}^\infty)\mathbf{X}||_2 \\
&= \sqrt{\sum_{j=1}^{n} ((e_i \hat{\mathbf{A}}^k)_j - (e_i \hat{\mathbf{A}}^\infty)_j)^2 \mathbf{X}_j^2} \\
&\leq \sqrt{\lambda_2^{2k} \frac{\sum\limits_{j=1}^{n} \tilde{d}_j \sum\limits_{p=1}^{f} \mathbf{X}_{jp}^2}{\tilde{d}_i}} = \lambda_2^k \sqrt{\frac{\sum\limits_{j=1}^{n} (\tilde{d}_j ||\mathbf{X}_j||_2^2)}{\tilde{d}_i}},
\end{aligned}
\tag{10}
$$

where $\mathbf{X}_{jp}$ denotes the $p^{th}$ feature of node $j$.

$\square$

Based on Lemma A.2 we then consider the smoothing distance for weighted averaged embedding features to the station state.

$$||[\hat{\mathbf{A}}^k \mathbf{X}]_i - [\hat{\mathbf{A}}^\infty \mathbf{X}]_i||_2 \tag{11}$$

Therefore there holds Theorem A.1

$$D_i(k) \leq \lambda_2^k \sqrt{\frac{\sum\limits_{j=1}^{n} (\tilde{d}_j ||\mathbf{X}_j||_2^2)}{\tilde{d}_i}}. \tag{12}$$

We denote the constant $\sum_{j=1}^{n}(\tilde{d}_j||\mathbf{X}_j||_2^2)$ as $cdx$ because it is independent with $j$, then Theorem A.1 can be written as

$$D_i(k) \leq \lambda_2^k \sqrt{\frac{cdx}{\tilde{d}_i}}. \tag{13}$$

We then analyze the factors affecting the Smoothing Weight on a specific node $v_i$. From Eq. 13 we know that the nodes with smaller degrees may have larger $D_i(k)$. Combined with definition 4.1, we infer that larger $D_i(k)$ makes the $\max_k D_i(k)$ more dominant after the softmax operation, causing that weighted average results depend more on itself and its near neighbors. Inversely, for the nodes with smaller degrees, its result of weighted average depends more equally on all itself, its near neighbors, and its distant neighbors.

At the same time, the Smoothing Weight of the node in a sparser graph ($\lambda_2$ is positively relative with the sparsity of a graph) decays slower as k increases. Thus, the weighted average result depends more equally on itself, its near neighbors and its distant neighbors. While for the nodes in a denser graph, the weighted average result depends more on its near neighbors and itself.

**Prevent over-smoothing.** Based on Theorem A.1 we then consider the smoothing distance for weighted averaged embedding features to the station state:

$$||\sum_{k=0}^{K}\omega_i(k) * [\hat{\mathbf{A}}^k\mathbf{X}]_i - [\hat{\mathbf{A}}^\infty\mathbf{X}]_i||_2 = ||\sum_{k=0}^{K}\omega_i(k) * ([\hat{\mathbf{A}}^k\mathbf{X}]_i - [\hat{\mathbf{A}}^\infty\mathbf{X}]_i)||_2$$

$$\leq \sum_{k=0}^{K}\omega_i(k) * ||[\hat{\mathbf{A}}^k\mathbf{X}]_i - [\hat{\mathbf{A}}^\infty\mathbf{X}]_i||_2 \tag{14}$$

$$= \sum_{k=0}^{K}\omega_i(k) * D_i(k).$$

When $\omega_i(k) = 1/(K+1)$, which means the average option is non-weighted, we have

$$\lim_{K\to\infty}\sum_{k=0}^{K}\omega_i(k) * D_i(k) = \lim_{K\to\infty}\frac{1}{K+1}\sum_{k=0}^{K}D_i(k)$$

$$\leq \lim_{K\to\infty}\frac{1}{K+1}\sum_{k=0}^{K}\lambda_2^k\sqrt{\frac{cdx}{\tilde{d}_i}} \tag{15}$$

$$= \lim_{K\to\infty}\frac{1}{K+1}\frac{1-\lambda_2^{K+1}}{1-\lambda_2}\sqrt{\frac{cdx}{\tilde{d}_i}}$$

$$= 0,$$

causing over-smoothing. When $\omega_i(k) = D_i(k)/(\sum_{l=0}^{K}D_i(l))$, let $\mathbf{X}_i^k = [\hat{\mathbf{A}}^k\mathbf{X}]_i - [\hat{\mathbf{A}}^\infty\mathbf{X}]_i$, we have:

$$||\hat{\mathbf{X}}_i - [\hat{\mathbf{A}}^\infty\mathbf{X}]_i||_2 = \lim_{K\to\infty}\omega_i(0)||\mathbf{X}_i^0 + \sum_{k=1}^{K}\frac{\omega_i(k)}{\omega_i(0)}\mathbf{X}_i^k||_2 \tag{16}$$

In the real-world graphs, nodes have different initial features, thus there is little chance that the combination of a node's neighboring features both lies in the opposite direction and is of the same norm of the node's initial feature. Suppose that there exits a constant $\epsilon > 0$ satisfying $\min\left(D_i(0), ||\mathbf{X}_i^0 + \sum_{k=1}^{K}\frac{\omega_i(k)}{\omega_i(0)}\mathbf{X}_i^k||_2\right) \geq \epsilon$ for node $i$, we have:

$$||\hat{\mathbf{X}}_i - [\hat{\mathbf{A}}^\infty\mathbf{X}]_i||_2 \geq \frac{\epsilon^2}{\lim_{K\to\infty}\sum_{k=0}^{K}D_i(k)} \geq \frac{\epsilon^2}{\frac{1}{1-\lambda_2}\sqrt{\frac{cdx}{\tilde{d}_i}}} > 0. \tag{17}$$

Thus we see that even $K$ goes to infinity, we are able to prevent the node representations from reaching the stationary state (the distance bound depends on the node degree $d_i$ and the initial feature

Table 3: Overview of the datasets.

| Dataset | Nodes | #Features | #Edges | #Classes |
|---------|-------|-----------|--------|----------|
| Cora | 2,708 | 1,433 | 5,429 | 7 |
| Citeseer | 3,327 | 3,703 | 4,732 | 6 |
| PubMed | 19,717 | 500 | 44,338 | 3 |
| Wiki | 2,405 | 4,973 | 17,981 | 17 |

$\mathbf{X}_i$). Note that in practice the representation at $k$-th smoothing step $\mathbf{X}^{(k)}$ achieves the stationary state much earlier than the infinity-th smoothing step we use in the theoretical analysis, so we use the softmax normalization in Equation 5 to produce a slightly larger bias towards features with longer distances to the stationary state.

**Node-adaptive weighting.** The above analysis *theoretically proved* our weighting scheme is able to prevent over-smoothing as the smoothing step goes to infinity. We now show that another advantage of our method is that it can fully leverage the multiple features over different smooth steps in a node-adaptive manner, which is different from the traditional routine of the smooth-blind and fixed scheme. Suppose that the feature of $k$ step is not over-smoothed yet for a specific node $i$, i.e., the distance of $D_i(k) = \epsilon_i > 0$, we have:

$$\omega_i(k) = \frac{\epsilon_i}{\lim_{K \to \infty} \sum_{k=0}^{K} D_i(k)} \geq \frac{\epsilon_i}{\frac{1}{1-\lambda_2} \sqrt{\frac{cdx}{\tilde{d}_i}}} > 0. \tag{18}$$

We see that as long as the feature is not over-smoothed, our method will assign a non-zero weight to the feature. Further, we see that the bound of weight can be affected by the over-smoothing distance of $\epsilon_i$ and degree $d_i$ of a specific node, implying an adaptive weighting strategy.

## B  DETAILS ON THE EXPERIMENTS

### B.1  DATASETS DESCRIPTION

Cora, Citeseer, PubMed, and Wiki are four popular network datasets. The first three (Kipf & Welling, 2016b) are citation networks where nodes stand for research papers and an edge exists between a node pair if one cites the other. Wiki (Yang et al., 2015) is a webpage network where nodes stand for webpages and an edge exists between a node pair if one links the other. Table 3 presents an overview of these four datasets.

### B.2  EXPERIMENTAL ENVIRONMENT

The experiments are conducted on a machine with Intel(R) Xeon(R) Gold 5120 CPU @ 2.20GHz, and a single NVIDIA TITAN RTX GPU with 24GB GPU memory. The operating system of the machine is Ubuntu 16.04. For software versions, we use Python 3.6, Pytorch 1.7.1, and CUDA 10.1.

## C  ADDITIONAL EMPIRICAL RESULTS

### C.1  PERFORMANCE ON THE NODE CLASSIFICATION TASK

In this subsection, we further assess the quality of the node embeddings generated by NAFS by the evaluation on the node classification task. We follow the linear evaluation protocol, which applies a linear classifier (i.e., Logistic Regression) to the node embeddings to generate final prediction results. We choose GCN (Kipf & Welling, 2016a), JK-Net (Xu et al., 2018), C&S (Huang et al., 2020), SGC (Wu et al., 2019), GAT (Veličković et al., 2017), PPRGo (Bojchevski et al., 2020), APPNP (Klicpera et al., 2018), and DAGNN (Liu et al., 2020) as comparison baselines on the node classification task. Note that C&S in our evaluation adopts MLP as the base model. The evaluation results on three popular datasets, Cora, Citeseer, and PubMed, are provided in Table 4.

The table shows that our method achieves comparable predictive accuracy as one of the state-of-the-art methods DAGNN. However, since the node embeddings are generated before training, our method

Table 4: Test accuracy on the node classification task.

| Methods | Cora | Citeseer | PubMed |
|---------|------|----------|--------|
| GCN | 81.8±0.5 | 70.8±0.5 | 79.3±0.6 |
| JK-Net | 81.9±0.4 | 70.7±0.7 | 78.8±0.7 |
| C&S | 76.7±0.4 | 70.8±0.6 | 76.5±0.5 |
| SGC | 81.0±0.2 | 71.3±0.5 | 78.9±0.5 |
| GAT | 83.0±0.7 | 72.5±0.6 | 79.0±0.3 |
| PPRGo | 82.4±0.3 | 71.3±0.5 | 80.0±0.4 |
| APPNP | 83.3±0.5 | 71.8±0.5 | 79.7±0.3 |
| DAGNN | **84.4±0.6** | 73.3±0.6 | **80.5±0.5** |
| NAFS-mean | 84.2±0.6 | 73.2±0.5 | 80.5±0.5 |
| NAFS-max | 83.8±0.7 | 73.4±0.6 | 80.4±0.6 |
| NAFS-concat | 84.1±0.6 | **73.5±0.4** | 80.3±0.4 |

Table 5: Scalablity comparison on sampled real-world graphs with different sample sizes.

| Methods | 10,000 | 20,000 | 30,000 | 50,000 | 100,000 |
|---------|--------|--------|--------|--------|---------|
| GAE | 10.2s | 36.3s | OOM | OOM | OOM |
| AGE | 68.5s | 256.2s | 727.7s | 2315.7s | OOM |
| NAFS-mean | 2.8s | 6.5s | 10.8s | 13.8s | 23.7s |

avoids performing recursive feature smoothing at each training epoch and storing the entire adjacency matrix on GPU. In this way, our method is more scalable and efficient to apply on large graphs than most state-of-the-art methods like DAGNN.

## C.2 SCALABILITY COMPARISON ON SYNTHETIC GRAPHS

A challenging task in today's graph learning is how to get decent node embedding on large graphs. To test the scalability of our proposed NAFS, we use the Erdős-Rényi graph generator in the Python package NetworkX (Hagberg et al., 2008) to generate artificial graphs of different sizes. The node sizes of the generated artificial graphs vary from 5,000 to 100,000, and the probability of an edge exists between two nodes is set to 0.0001. The experiment settings are altered compared to Sec. 6.4: both GAE and AGE are trained on an NVIDIA TITAN RTX which has 24 GB of memory, since only using CPU to train being unacceptable on large graphs in real-world applications.

The overall experiment results are shown in Figure 6. Figure 6(a) shows a more detailed comparison on relatively small artificial graphs whose node sizes range from 5,000 to 20,000. Besides, Figure 6(b) shows that GAE encounters the out-of-memory problem on the artificial graph composed of 30,000 nodes, while AGE encounters the out-of-memory problem at graph size 80,000. Our proposed NAFS is successfully carried out on all the artificial graphs. On the artificial graph consists of 100,000 nodes, NAFS accomplishes the node embedding generation in 66.1 seconds, which is less than the running time of AGE on the artificial graph consists of 10,000 nodes, 80.9 seconds. The graph size limit of our proposed NAFS is bounded by the CPU memory size. As long as one can successfully execute the multiplication of the sparse adjacency matrix and the feature matrix, NAFS can then be implemented.

## C.3 SCALABILITY COMPARISON ON REAL-WORLD GRAPHS.

To verify the scalability advantage of NAFS, we also add the evaluation on real-world datasets. Concretely, we sample subgraphs of different scales from the full ogbn-products graphs via uniformly random node selection and then report the sampled graph size (i.e., number of nodes) and the corresponding runtime (seconds) in Table 5. The experimental results show that NAFS can support

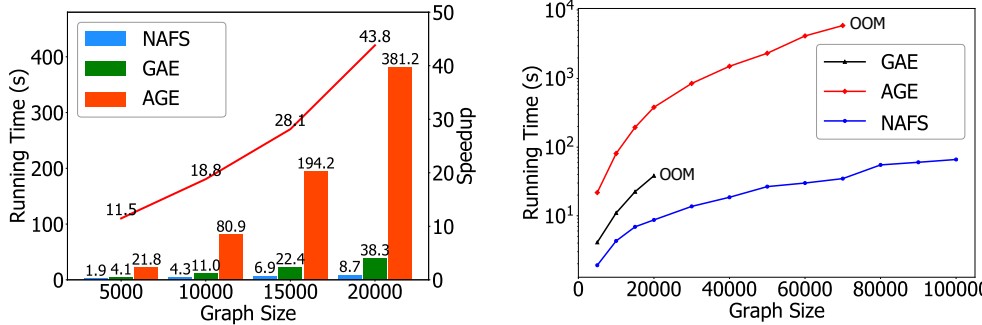

Figure 6: Scalablity comparison on synthetic graphs, OOM is "out of memory".

larger graphs (i.e., larger than 30,000 nodes). Besides, it is significantly faster than the compared baselines, especially for large graph datasets.

## C.4 VISUALIZATION OF EMBEDDING

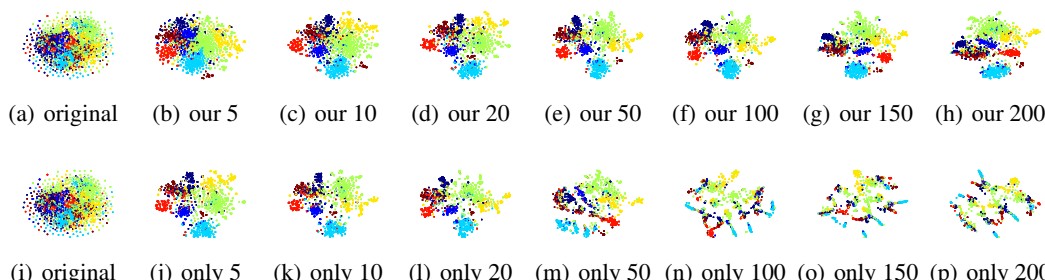

Figure 7: T-SNE visualization of node embedding on Cora.

To better understand why NAFS is effective, we visualize the node embedding generated by NAFS using T-SNE (Van der Maaten & Hinton, 2008) on the Cora dataset. Moreover, we also visualize the node embedding of $\hat{\mathbf{A}}^k \mathbf{X}$ for reference. All the visualization results are shown in Figure 7. The first row is the results of our proposed NAFS, and the second row is the results of $\hat{\mathbf{A}}^k \mathbf{X}$.

Figure 7(c) and 7(k) illustrate that at hop 10, the node embedding produced by NAFS and $\hat{\mathbf{A}}^k \mathbf{X}$ are both distinguishable. But as the value of maximal smoothing step becomes larger, the node embedding of $\hat{\mathbf{A}}^k \mathbf{X}$ falls into total disorder, like the situation showed by Figure. 7(n), 7(o) and 7(p). At the same time, NAFS is able to maintain the distinguishable results even when the value of the maximal number of smoothing step increases to 200.

## D MORE DETAILS OF NAFS

### D.1 PSEUDO CODE OF NAFS

Alg. 1 shows the whole pipeline of our proposed NAFS. We first initialize $\mathbf{X}^{(0)}$ as the original feature matrix $\mathbf{X}$. Given the normalization parameter $r_t$, we obtained the corresponding normalized adjacency matrix $\hat{\mathbf{A}}_{r_t} = \widetilde{\mathbf{D}}^{r_t-1} \widetilde{\mathbf{A}} \widetilde{\mathbf{D}}^{-r_t}$, which acts as knowledge extractors (line 4). After that, for each node, we use E.q. 4 to calculate the Over-smoothing Distance with all the $k$ ranging from 0 to $K$ (line 6, 7). Then we calculate its Aggregation Weights through Eq. 5 (line 8, 9). After obtaining Aggregation Weights with all $k$ and $i$, we construct the Aggregation Weight matrix for each $k$ with

---

**Algorithm 1: NAFS pipeline.**

---

**Input:** Smoothing step $K$, feature matrix $\mathbf{X}$, adjacency matrix $\mathbf{A}$, and $\{r_1, r_2, ..., r_T\}$.
**Output:** Graph embedding matrix $\mathbf{Z}$.

1   Initialize the feature matrix $\mathbf{X}^{(0)} = \mathbf{X}$;

2   **Operation 1: Feature Smoothing**

3   **for** $1 \leq t \leq T$ **do**

4      Update the normalized adjacency matrix $\hat{\mathbf{A}}_{r_t} = \widetilde{\mathbf{D}}^{r_t - 1} \widetilde{\mathbf{A}} \widetilde{\mathbf{D}}^{-r_t}$ ;

5      **for** $1 \leq i \leq n$ **do**

6         **for** $0 \leq k \leq K$ **do**

7            Calculate $D_i(k)$ and $w_i(k)$ with Eq. 4 and 5, respectively;

8      **for** $0 \leq k \leq K$ **do**

9         Construct $\mathbf{W}(k)$ with Eq. 6;

10      Smooth the node features $\mathbf{X}$ with $\hat{\mathbf{X}}^{(t)} = \sum\limits_{k=0}^{K} \mathbf{W}(k) \hat{\mathbf{A}}_{r_t}^k \mathbf{X}$;

11   **Operation 2: Feature Ensemble**

12   Compute the final embedding $\mathbf{Z}$ with $\mathbf{Z} \leftarrow \oplus_{i \in \{1,2,...,T\}} \hat{\mathbf{X}}^{(i)}$.

---

Eq. 6 (line 10, 11). Next, we compute the NAFS output $\hat{\mathbf{X}}^{(t)}$ with $\hat{\mathbf{X}}^{(t)} = \sum\limits_{k=0}^{K} \mathbf{W}(k) \hat{\mathbf{A}}_{r_t}^k \mathbf{X}$ (line 12).

Finally, we compute the final embedding result of all $t$ through $\mathbf{Z} \leftarrow \oplus_{i \in \{1,2,...,T\}} \hat{\mathbf{X}}^{(i)}$ (line 14).

### D.2   MOTIVATION OF FEATURE ENSEMBLE

Adopting different smoothing operators in the feature smoothing operation (the normalized adjacency matrix $\hat{\mathbf{A}}$ in Eq. 1) is equivalent to smoothing features in different manners. Other than the normalized adjacency matrix $\hat{\mathbf{A}}$, there are many alternatives that have been proposed recently. For example, GraphSAGE (Hamilton et al., 2017) designs three smoothing operators (i.e., Mean, LSTM and Pooling) to flexibly capture the information of neighboring nodes. SIGN (Frasca et al., 2020) enriches the smoothing operators with Personalized-PageRank-based (Klicpera et al., 2019) and triangle-based (Monti et al., 2018) adjacency matrices. However, these methods are not designed for graph representation learning and different smoothing operators may results in diverse smoothed features. For better node representation, feature ensemble is used to combine the smoothed feature under different smoothing operators.

### D.3   RELATIONS WITH OTHER SCALABLE GNN ARCHITECTURES

The scalable GNNs can be roughly classified into two categories: (a) "first propagate then predict"; (b) sampling + ordinary GNN. The first category includes famous GNN models like SGC (Wu et al., 2019) and SIGN (Frasca et al., 2020). They disentangles the coupled propagation and transformation operations in traditional GCN layers (Kipf & Welling, 2016a), and execute all the propagation operations as preprocessing. The most representative GNN model of the second category is GraphSAGE (Hamilton et al., 2017), and many other works have been proposed following it, like FastGCN (Chen et al., 2018) and GraphSAINT (Zeng et al., 2019). The main contribution of these works are effective sampling strategies that can preserve the most valuable information from the original graph. The sampling strategies always act as a plug-and-play module, and can be combined with ordinary GNNs, which allows the latter to perform on large-scale graphs. GNNs belong to the "first propagate then predict" category usually enjoy higher efficiency since they avoid recursively performing propagation in each training epoch. Our proposed NAFS belongs to the "first propagate then predict" category.

### D.4   ADVANTAGE IN DISTRIBUTED SETTINGS

NAFS can also be adapted to the distributed environment. The feature smoothing process in NAFS are sparse matrix dense matrix multiplications, which have mature implementations in distributed environments. Besides, this process only need to be pre-computed at once. In contrast, during

each training iteration of GAE and its variants, each node needs to repeatedly pull the intermediate representations of other nodes, which leads to high communication cost.

