# OpenReview forum: "NAFS: A Simple yet Tough-to-Beat Baseline for Graph Representation Learning"
_ICLR.cc/2022/Conference — ICLR 2022 Submitted_

### Official Review · Reviewer_YM4P · 2021-10-29

**Correctness:** 4
**Technical Novelty And Significance:** 2
**Empirical Novelty And Significance:** 2
**Recommendation:** 5
**Confidence:** 4

**Main Review:**

**Strengths:**
1. Simple approach that seems to work well
2. Well-presented and easy to read
3. Rather large set of baseline methods used

**Weaknesses**
1. Old, small-scale datasets
2. No error bars/standard deviations were reported
3. Scalability experiments only carried out on synthetic graphs

**Suggestion**
1. Evaluate the method on more large-scale, modern datasets, e.g. OGB datasets or graphlearning.io
2. Perform scalability on real-world datasets
3. Discuss "linear" GNN architectures, e.g., https://arxiv.org/abs/1810.05997
4. Discuss other scalable GNN alternatives, e.g., ones based on label propagation, see https://arxiv.org/abs/2010.13993

**Comments**
1.  The discussion in section 2.2 only applies to a specific GNN layer, namely GCN

**Question**
1. Is your approach also liftable to other GNN architectures besides GCN?



**Summary Of The Paper:**

The paper deals with (unsupervised) learning with graphs, specifically node-level tasks. Inspired by spectral GNNs, specifically Graph Convolutional Networks (GCN), the authors propose a simple neighborhood smoothing technique to capture the graph structure around each node in the given graph. Contrary to GNN, the proposed algorithm is parameter-free and hence scales better than their end-to-end trained GNN counterpart.

Motivated by the problem of over-smoothing of GCN (Li et al., 2018), they propose the so-called "over-smoothing distance" measuring how close a node's feature is to be over-smoothed, which is simply the row-wise distance between $A^k X$ and $A^{\infty}X$ with regard to the Euclidian distance, see Def. 3.1. Based on this distance they define the smoothing weight matrix which is then used to weight the neighboring node features during neighborhood aggregation. The resulting features $\hat{X}$ are computed as $\hat{X} =  \sum^{K}_{k=0} W(k)\hat{A}X$, where $W(k)$ is the $k$th smoothing weight matrix.

The proposed architecture is evaluated on standard, old, small-scale (unsupervsied) link and node classification (Cora, Citeseer, PubMed) tasks showing somewhat classification performance while showing a considerable speedup in computation time.

**Summary Of The Review:**

The approach is very simple, it offers no significant theoretical or methodological contributions, and the experimental study is not executed well enough, i.e., it only uses small-scale, old or synthetic datasets. Further, the are some problems in the experimental protocol, e.g., no standard deviations are reported.

---

> ### Author Response · Authors · 2021-11-18
> **Response to Reviewer 3, Part 1**
>
> Thanks for your insightful review. We appreciate your assessment about this paper being "well-presented and easy to read" and our method being a "simple approach that seems to work well". The answers to your concerns are as follows.
>
> ### Q1: Evaluation on more large-scale, modern datasets.
>
> In fact, we have attempted to evaluate the baseline methods on more large-scale OGB datasets. Although our method still works when the graphs grow larger, we find that most GNN-based unsupervised graph embedding baselines can not be directly executed on large-scale datasets. The main reason is that they have to reconstruct the adjacency matrix, which leads to extremely large memory costs when training on GPUs. For example, the smallest OGB dataset contains 235k nodes, which is larger than PubMed dataset used in our experiments. It requires about 205GB GPU memory to store the dense-form reconstructed adjacency matrix in the link prediction tasks. Therefore, for **unsupervised** link prediction and node clustering tasks, it's difficult to evaluate our method on large-scale graphs due to the lack of scalable GNN-based graph embedding baselines.
>
> Fortunately, for **supervised** tasks such as node classification, we find that many semi-supervised GNN methods work on large-scale graphs. So we evaluate the quality of our embeddings on more large-scale, modern datasets. To this end, we feed the node embeddings generated by our method into a linear classifier (i.e., Logistic Regression) to evaluate its performance on the node classification task. The predictive accuracy of our method is compared with several semi-supervised GNN methods. We conduct our evaluation on the large-scale ogbn-arxiv dataset, and the results are as follows:
>
> | ogbn-arxiv    | MLP              | MLP+C\&S         | GCN              | SGC              | SIGN             | DAGNN            | JK-Net           | APPNP            | GBP              | GAT              | NAFS-mean        |
> | ------------- | ---------------- | ---------------- | ---------------- | ---------------- | ---------------- | ---------------- | ---------------- | ---------------- | ---------------- | ---------------- | ---------------- |
> | Test Accuracy | 55.50 $\pm$ 0.23 | 71.58 $\pm$ 0.18 | 71.74 $\pm$ 0.29 | 71.72 $\pm$ 0.35 | 71.95 $\pm$ 0.14 | 72.09 $\pm$ 0.25 | 72.19 $\pm$ 0.21 | 72.34 $\pm$ 0.32 | 72.45 $\pm$ 0.17 | 73.56 $\pm$ 0.21 | 73.02 $\pm$ 0.25 |
> | Runtime       | 1.842s           | 8.756s           | 25.228s          | 5.288s           | 10.956s          | 58.463s          | 52.614s          | 45.524s          | 17.075s          | 147.692s         | 22.221s          |
>
> The table shows that our method (NAFS-mean) outperforms many competitive baselines on the large-scale graph dataset ogbn-arxiv. Although the test accuracy of GAT is slightly higher than NAFS-mean, our method achieves over 6x speedup than GAT.
>
>
>
> ### Q2: No error bars/standard deviations.
> As suggested by the reviewer, we have added the standard deviations in the experiment results. Here we want to mention that the performance of our method is deterministic (i.e., the error is zero) due to the training-free manner. Concretely, we apply the K-Means for node clustering and the inner-product decoder for link prediction. Therefore, the performance of our method is deterministic if the maximal smoothing step and smoothing operations are defined.
>
>
> ### Q3: Scalability comparison on real-world datasets.
> Based on the reviewer's suggestion, we have added the scalability experiments on real-world datasets. Concretely, we sample subgraphs of different scales from the full ogbn-products graphs via uniformly random node selection and then report the sampled graph size (i.e., number of nodes) and the corresponding runtime (seconds) in Table 5. The experimental results show that NAFS can support larger graphs (i.e., larger than 30,000 nodes). Besides, it is significantly faster than the compared baselines, especially for large graph datasets.
>
> **Table 5: Scalablity comparison on sampled real-world graphs with different sample sizes.**
>
> | Method    | 10,000 | 20,000 | 30,000 | 50,000  | 100,000 |
> | --------- | ------ | ------ | ------ | ------- | ------- |
> | GAE       | 10.2s  | 36.3s  | OOM    | OOM     | OOM     |
> | AGE       | 68.5s  | 256.2s | 727.7s | 2315.7s | OOM     |
> | NAFS-mean | 2.8s   | 6.5s   | 10.8s  | 13.8s   | 23.7s   |
>
> We have added the experimental results and the corresponding analysis in Appendix C.3.

---

> > ### Author Response · Authors · 2021-11-18
> > **Response to Reviewer 3, Part 2**
> >
> > ### Q4: Discussions about Linear GNNs \& Scalable GNNs.
> > Thanks for reminding us of the related "linear" GNN architecture like APPNP and scalable GNN alternatives like Correct and Smooth (C\&S). We identify the following important differences between our method and these GNN alternatives.
> >
> > First, our method is an unsupervised graph embedding method, whereas these GNN alternatives such as APPNP and C\&S focus on the semi-supervised setting (i.e., node classification) tasks.
> > Second, APPNP and C\&S adopt a propagation scheme based on Personalized PageRank (PPR), which always maintains certain input information to slow down the occurrence of over-smoothing. The expected aggregation weights are controlled by the same restart probability $\alpha$. Therefore, these GNN alternatives still fall into the routine of aggregating information from multi-hop neighbors in a fixed manner. Instead, due to the observation that different nodes have different convergence speeds to the over-smoothing state (see our theorem A.1), our method employs a distance-aware weighting scheme that aggregates information from multi-hop neighbors in a node-specific manner.
> >
> > As suggested by the reviewer, we have added more discussions about the "linear" GNN architectures in Section 3.1 and the experiment to compare our method with APPNP and C&S under the supervised task of node classification in Table 4 in Appendix C.1 of the revised manuscript. Parts of the results are as follows:
> >
> > | Methods   | Cora           | Citeseer       | PubMed         |
> > | --------- | -------------- | -------------- | -------------- |
> > | APPNP     | 83.3 $\pm$ 0.5 | 71.8 $\pm$ 0.5 | 79.7 $\pm$ 0.3 |
> > | C&S       | 76.7 $\pm$ 0.4 | 70.8 $\pm$ 0.6 | 76.5 $\pm$ 0.5 |
> > | NAFS-mean | 84.2 $\pm$ 0.6 | 73.2 $\pm$ 0.5 | 80.5 $\pm$ 0.5 |
> >
> >
> >
> >
> > ### Q5: Liftable to other GNNs.
> > The reviewer raises a very good point. Although we utilize the augmented normalized adjacency like GCNs in the original manuscript, our method can be easily extended to incorporate other propagation procedures. Here, we take the Personalized PageRank (PPR) used in APPNP as an example. To incorporate the PPR, We can adopt the same weighting strategy to combine multiple smoothed features under different values of restart probability $\alpha$, like $0.1, 0.2, \ldots, 0.9$. And for each node, we set the weight as the distance from the smoothed node feature to the stationarity given by $\hat{\mathbf{A}}^{\infty}\mathbf{X}$.

---

> > > ### Comment · Reviewer_YM4P · 2021-11-22
> > > **Answer**
> > >
> > > I read the rebuttal and slightly increased my score.

---

> > > > ### Author Response · Authors · 2021-11-23
> > > > **Answer for theoretical contributions to Reviewer 3**
> > > >
> > > > We really appreciate the reviewer for increasing the score for this paper.
> > > > To address the problem that " it offers no significant theoretical or methodological contributions" as suggested by the reviewer, we have improved the theoretical and methodological significance of this work in Appendix. A of our revised manuscript.
> > > >
> > > > Concretely, we propose a new methodological principle to combine representations over different smoothing steps: Assigning weights proportional to the over-smoothing distance of the features at each smoothing step $k$, i.e., $\omega_i(k) \propto D_i(k)$. It is worth pointing out that the speed to approach the over-smoothing stationarity is *heterogeneous* across nodes (as implied by our theorem A.1 in Appendix. A). Therefore, the methodological contribution lies in a new **over-smoothing-distance-aware** and **node-adaptive** weight assignment scheme, which is fundamentally different from the traditional routine of the smooth-blind and fixed scheme.
> > > >
> > > > A key theoretical contribution is that the proposed methodological principle is **theoretically proved** to prevent
> > > > over-smoothing as the smoothing step goes to infinity. By contrast, the final representation will be over-smoothed if only adopting fixed and uniform weights. The Appendix. A in the revised manuscript provides the detailed theoretical analysis.
> > > >
> > > > Here, we provide a **theoretical analysis sketch** to understand how our weighting principle actually impacts final representations from a theoretical perspective.
> > > > We first give an upper bound to the over-smoothing distance at each step $k$, showing $D_{i}(k) \to 0$ quickly as $k$ increases.
> > > > Next, we show simply setting the weight uniformly as $\omega_i(k) = 1/(K+1)$ would over-smooth the final representation $\hat{\mathbf{X}}_i$ of a node $i$ ( i.e., $||\hat{\mathbf{X}}\_i-[\hat{\mathbf{A}}^{\infty}\mathbf{X}]\_i||\_{2} \to 0$), because the uniform weight would make over-smoothed features dominate the final representation. By contrast, we show that distance-aware weighting scheme actually prevents the final node representations from converging to the identical one (i.e., there is a constant $\epsilon > 0$ making
> > > > $||\hat{\mathbf{X}}\_i-[\hat{\mathbf{A}}^{\infty}\mathbf{X}]\_i||\_{2} > \epsilon$). This is because the weight decreases at the same rate with over-smoothing distance of smoothed features, excluding over-smoothed features from the final representation while retaining all feature that are not over-smoothed (i.e., $\omega\_i(k)>0$ given $D\_i(k) > 0$).
> > > >
> > > > The empirical results in the original manuscript further demonstrate the validity of our principle as it could fully leverage features at a different level of localities in a node-adaptive manner.
> > > >
> > > > We hope this answer could address your concern about the theoretical or methodological contributions, and we are happy to respond if you have additional comments or concerns on our responses.
> > > >
> > > > Respectfully,
> > > >
> > > > Paper608 Authors

---

> > > > > ### Author Response · Authors · 2021-11-27
> > > > > **Response to Reviewer 3**
> > > > >
> > > > > Thanks for your helpful and valuable reviews! We particularly appreciate the advice of
> > > > > 1) Adding more theoretical analysis;
> > > > > 2) Evaluation on large OGB datasets;
> > > > > 3) Scalability analysis on real-world datasets;
> > > > > 4) Discussions about Linear GNNs & Scalable GNN;
> > > > > 5) Discussions about the liftability to other GNNs.
> > > > >
> > > > > We have carefully responded to the above advice, and we hope to address all your points.
> > > > >
> > > > > We are happy to respond if you have additional comments or concerns on our responses.
> > > > >
> > > > > Respectfully,
> > > > >
> > > > > Paper608 Authors

---

### Official Review · Reviewer_w2Qg · 2021-11-03

**Correctness:** 4
**Technical Novelty And Significance:** 3
**Empirical Novelty And Significance:** 3
**Recommendation:** 6
**Confidence:** 4

**Main Review:**

The authors took a novel perspective and presented NAFS which does not explicitly require parameter learning. It is also a bold idea that, in addition to separate feature transformation from feature smoothing, the model removes feature transformation altogether.

A few questions:
1. Since the method has no feature transformation step, the construction of node feature in the initial step seems to become rather important. What are the features used for each of the dataset in the experiment (Cora, Citeseer, PubMed, Wiki)? I would be good to state them clearly in the paper.

2. Would be useful to comment on what the authors think might be the limitation of this method. For example, NAFS by design does not require task specific training, although hyper-parameter tuning allow the model to serve for different tasks. Then what types of graphs/tasks might work best with this model, or is it indifferent.


**Summary Of The Paper:**

The authors of this paper took a novel perspective to present the node-adaptive feature smoothing (NAFS) algorithm, which generates node embeddings without explicit training/parameter learning. The method first performs feature smoothing, then combines the smoothed features using adaptive weights which are node-specific. They further enhanced this method by ensembling the smoothed features extracted with different hyper-parameters. The authors have conducted many experiments to validate the model performance, and demonstrate the model's efficiency empirically.

**Summary Of The Review:**

In summary, the paper is well presented. The motivations and the authors' insights to this model is well explained. The method is clearly described. The authors performed many experiments to validate the model's performance, with additional ablation studies. Would be nice to have more discussion on which scenarios the model works better/worse.

---

> ### Author Response · Authors · 2021-11-18
> **Response to Reviewer 2**
>
> We thank the reviewer for your reviews. Please find our answers below:
>
> ### Q1: Features used for each of the dataset.
> Due to the length limitation of the main paper, we provided the dataset description in detail in Appendix B.1 of the original manuscript, which says, "In Cora and Citeseer, the node features are binary word vectors; while in PubMed and Wiki, each node has a TF-IDF weighted word vector".
>
> We appreciate your suggestion, and have added the desciption in Section 6.1 of the revised manuscript.
>
>
> ### Q2: Limitation of this method.
> Thanks for your suggestion on discussing what types of graphs/tasks might work best with this model. Our method works well on tasks/graphs where large smoothing steps are required to collect neighborhood information for each node. In terms of graph type, our method performs better on the sparse graphs with low edge density since we can exploit distant neighborhood information without the over-smoothing problem. For example, in the link predication task, our method provides higher performance gain on the "PubMed" dataset, which has the lowest edge density among the three datasets. In terms of task type, our method is more suitable for those who require more global information. For example, our method provides a higher performance gain on graph clustering tasks than link prediction tasks.

---

> > ### Author Response · Authors · 2021-11-27
> > **Response to "which scenarios the model works better/worse"**
> >
> > Thanks for your valuable reviews! We have added the following experiments to better answer your question about "what types of graphs/tasks might work best with this model, or is it indifferent".
> >
> > Our method works well on tasks/graphs where large smoothing steps are required to collect neighborhood information for each node. In other words, our method performs better on the sparse graphs with low feature and edge sparsity. While existing methods may lead to over-smoothing (Equation 3 in Section 3.2), NAFS **explicitly** avoids this issue by aggregating the neighborhood information of different steps based on the Smoothing Weight.
> >
> > To verify the above conclusion, we conduct experiments on both the unsupervised link prediction task and supervised node classification task to evaluate the performance of NAFS under two sparsity settings: edge sparsity and feature sparsity.
> >
> > **Experimental Settings.** Under the edge sparsity setting, we randomly remove some edges in the original PubMed dataset to strengthen the edge sparsity issue. The edges removed from the original graph are kept the same across all the compared methods under the same edge removing rate. Under the feature sparsity setting, we randomly choose some nodes in the original PubMed dataset and set their feature vectors to all-zero vectors. The selected nodes are kept the same across all the compared methods under the same feature removing rate.
> >
> > We report the performance of GAE and NAFS-mean on the link prediction task and the performance of GCN and NAFS-mean on the node classification task, where a Logistic Regression classifier is applied to the embeddings generated by NAFS-mean to produce its predictions on node classes. The evaluations are conducted under the edge removing rate and the feature removing rate being set to 0.2, 0.4, 0.6, respectively. We repeat each baseline method ten times and report the mean performance and the standard deviations in the following six tables.
> >
> > **Experimental Results.** The experimental results from the following tables show that 1) under both edge and feature sparsity settings, NAFS consistently outperforms GAE and GCN on the link prediction task and the node classification task, respectively. 2）the performance gains are larger in sparser graphs with larger feature/edge removing rates. Concretely, the AUC of NAFS outperforms GAE by a margin of 1.0\% if the edge is not removed, and the performance gain has increased to 3.7\% when the edge removing rate is 0.6.
> > Compared with baseline methods, NAFS can effectively exploit distant neighborhood information without the over-smoothing problem and thus get better performance in sparse graphs.
> >
> > **Performance under different levels of edge sparsity.**
> >
> > | AUC, edge  | 0.0            | 0.2            | 0.4            | 0.6            |
> > |:--------- | -------------- | -------------- | -------------- | -------------- |
> > | GAE       | 96.4 $\pm$ 0.4 | 93.4 $\pm$ 0.6 | 92.6 $\pm$ 0.5 | 90.6 $\pm$ 0.5 |
> > | NAFS-mean | 97.4 $\pm$ 0.0 | 96.9 $\pm$ 0.0 | 95.9 $\pm$ 0.0 | 94.3 $\pm$ 0.0 |
> >
> > | AP, edge   | 0.0            | 0.2            | 0.4            | 0.6            |
> > | --------- | -------------- | -------------- | -------------- | -------------- |
> > | GAE       | 96.5 $\pm$ 0.5 | 93.7 $\pm$ 0.4 | 92.4 $\pm$ 0.6 | 90.4 $\pm$ 0.5 |
> > | NAFS-mean | 97.2 $\pm$ 0.0 | 96.4 $\pm$ 0.0 | 95.2 $\pm$ 0.0 | 93.5 $\pm$ 0.0 |
> >
> > | Test ACC, edge  | 0.0            | 0.2            | 0.4            | 0.6            |
> > | --------- | -------------- | -------------- | -------------- | -------------- |
> > | GCN       | 79.3 $\pm$ 0.6 | 71.5 $\pm$ 0.5 | 67.1 $\pm$ 0.4 | 60.0 $\pm$ 0.6 |
> > | NAFS-mean | 80.5 $\pm$ 0.5 | 77.0 $\pm$ 0.6 | 74.8 $\pm$ 0.5 | 71.4 $\pm$ 0.5 |
> >
> > **Performance under different levels of feature sparsity.**
> >
> > | AUC, feature | 0.0            | 0.2            | 0.4            | 0.6            |
> > |:----------- | -------------- | -------------- | -------------- | -------------- |
> > | GAE         | 96.4 $\pm$ 0.4 | 89.5 $\pm$ 0.6 | 83.5 $\pm$ 0.5 | 77.4 $\pm$ 0.5 |
> > | NAFS-mean   | 97.4 $\pm$ 0.0 | 93.1 $\pm$ 0.0 | 87.7 $\pm$ 0.0 | 81.7 $\pm$ 0.0 |
> >
> > | AP, feature | 0.0            | 0.2            | 0.4            | 0.6            |
> > | ---------- | -------------- | -------------- | -------------- | -------------- |
> > | GAE        | 96.5 $\pm$ 0.5 | 90.4 $\pm$ 0.4 | 85.5 $\pm$ 0.6 | 80.9 $\pm$ 0.5 |
> > | NAFS-mean  | 97.2 $\pm$ 0.0 | 94.4 $\pm$ 0.0 | 91.0 $\pm$ 0.0 | 86.8 $\pm$ 0.0 |
> >
> > | Test ACC, feature | 0.0            | 0.2            | 0.4            | 0.6            |
> > | ----------- | -------------- | -------------- | -------------- | -------------- |
> > | GCN         | 79.3 $\pm$ 0.6 | 77.6 $\pm$ 0.5 | 76.4 $\pm$ 0.4 | 74.2 $\pm$ 0.6 |
> > | NAFS-mean   | 80.5 $\pm$ 0.5 | 79.8 $\pm$ 0.6 | 79.0 $\pm$ 0.5 | 77.1 $\pm$ 0.5 |
> >
> > We hope the above analysis can address your concern, and we are happy to respond if you have additional comments or concerns on our responses.
> >
> > Respectfully,
> >
> > Paper608 Authors

---

### Official Review · Reviewer_SHxg · 2021-11-06

**Correctness:** 3
**Technical Novelty And Significance:** 3
**Empirical Novelty And Significance:** 2
**Recommendation:** 6
**Confidence:** 4

**Main Review:**

The paper tackles the problem of over-smoothing by proposing a method to differently weight the smoothing for each node. The problem is well known and in my opinion methods to better understand and solve it are very relevant for the community.

**Positive aspects**

While the connection of over-smoothing with powers of the adjacency matrix is not novel per se, I did not know about prior methods tackling the problem as it has been done in this paper. However, I should highlight that just recently another **very** similar paper was published on arxiv (Zhang et Al. "Node Dependent Local Smoothing for Scalable Graph Learning" https://arxiv.org/abs/2110.14377 ). While it seems recent enough to not put the contribute of this paper into discussion, I think it would be worth citing it and comparing its method to the presented one.

The approach is presented quite clearly and in a well-structured way. I appreciate that the authors introduce it as a baseline: better results might be obtained by adding a learning component to it, but it is interesting to see how far adaptive smoothing alone can go.

**Main concerns and chances for improvement**

By being introduced as parameterless, NAFS is compared with other methods (autoencoders, adversarial models) which are typically unsupervised. Moreover, the only supervised task it is tested on is the link prediction one. I think further experiments are required to better understand how useful the method can be. In particular, I think it would be interesting to see how it performs on other supervised tasks such as node classification (e.g. suing the linear evaluation protocol, used to assess the performances of many recent graph SSL methods).

The weighting scheme looks intuitively useful, but I believe it would be useful to understand how it actually impacts final representations from a more theoretical perspective. How do weights change node representations from converging to the same identical one?

Few minor corrections:
- page 1: "and these methods share two major limitations" ("and" looks superfluous)
- page 2: $\hat{A} = A + I_n$ should be $\tilde{A} = A + I_n$
- the $r$ convolution coefficient which is introduced in Section 4.2 previously appears in Equation (1) and this makes the equation less clear. I think it would be easier for the reader to have its description close to (1) instead.


**Summary Of The Paper:**

The paper presents NAFS (Node-Adaptive Feature Smoothing), a method that constructs node representations by relying on smoothing only, i.e. without parameter learning. To do this, the authors first provide a formulation for the smoothing operator after infinite steps, i.e. when the stationary state is reached. They then define over-smoothing distance as a way to assess how much a node is close to the stationary state after k smoothing steps. Finally, they use over-smoothing distances to calculate a different smoothing weight for each node.

Experiments show that representations obtained by smoothing with these weights, together with feature ensembles obtained by applying different convolution coefficients, provide performances in clustering and link prediction tasks which are comparable, if not better, than many other state-of-the-art approaches.

**Summary Of The Review:**

The paper tackles an interesting problem in a clear and reasonable way. The main issues I see here are the evaluation, that to me still looks quite limited, and the lack of a theoretical interpretation for the weighting scheme.

While a possibly negative outcome of the evaluation of a node classification task does not concern me too much (the method itself is defined as a baseline and I think it does not require to be state-of-the-art in every single benchmark for it to be valuable), I think motivating the weighting scheme by formally demonstrating how it impacts smoothing when $K$ grows is an important step to make the paper more convincing.

----
I have read the authors' replies to the other reviewers and myself and they look convincing to me. I particularly appreciate they addressed my comments on related works, theoretical analysis, and further experiments on supervised tasks. They also took care of adding mean/std results for both new and old experiments (even when results are deterministic, which is great to emphasize). This makes the paper way more convincing in my opinion and worth being accepted, which is why I increased my score accordingly.

---

> ### Author Response · Authors · 2021-11-18
> **Response to Reviewer 1, Part 1**
>
> We appreciate your assessment about the proposed NAFS algorithm "is presented quite clearly and in a well-structured way". Thanks for your constructive feedback! We believe that addressing this feedback will make our paper significantly stronger.
>
> ### Q1: Discussion with NDLS.
> Thanks for reminding us of the recent related work -- NDLS. We have read the paper in detail, and here we point out two main differences between our method and NDLS:
>
> (a) While NDLS searches for the optimal smoothing step and generates node embeddings via simple average, our method highlights how to combine the propagated features and proposes a node-adaptive solution based on the Smoothing Weight (Section 4). The goal of our method and NDLS is orthogonal, yet it is quite interesting to study the combined effects of our method and NDLS in future work.
>
> (b) In our method, we design a feature ensemble operation which applies several propagation strategies (i.e., setting different values of $r$ in $\small \mathbf{\hat{A}}_r = \tilde{\mathbf{D}}^{r-1}\tilde{\mathbf{A}}\tilde{\mathbf{D}}^{-r}$) to make the best of the propagated features. However, NDLS and most other GNNs support only one propagation strategy, which would be less robust when tackling a variety of tasks and datasets. Figure 5(b) in the original manuscript demonstrates the superiority of the ensemble operation (Concat vs. r=0.3 only).
>
> (c\) Though both NDLS and our method focus on generating node embeddings, NDLS evaluates the quality of its embeddings only on supervised node classification tasks. Instead, we conduct extensive experiments under different settings in our method, including unsupervised node clustering and supervised link prediction. In addition, we have also added supervised node classification as suggested in Question 2.
>
> As suggested by the reviewer, we have cited NDLS and added the discussion in the revised manuscript.
>
>
> ### Q2: Other supervised tasks like node classification.
> We did not include many supervised tasks in the original manuscript because our method aims to generate high-quality node embeddings. To demonstrate the quality of node embeddings, we follow the settings of empirical studies in previous works [1][2][3], which adopt the node clustering and link prediction tasks as shown in Section 6 in the original manuscript.
>
> Yet, we agree that it's reasonable to further assess the performance of our method on more supervised tasks. To this end, we feed the node embeddings generated by our method into a linear classifier (i.e., Logistic Regression) to evaluate its performance on the node classification task. We include the results on three public datasets as follows:
> **Table 4: Test accuracy on the node classification task.**
>
> | Methods         | Cora               | Citeseer           | PubMed             |
> | --------------- | ------------------ | ------------------ | ------------------ |
> | GCN             | 81.8 $\pm$ 0.5     | 70.8 $\pm$ 0.5     | 79.3 $\pm$ 0.6     |
> | JK-Net          | 81.9 $\pm$ 0.4     | 70.7 $\pm$ 0.7     | 78.8 $\pm$ 0.7     |
> | C&S             | 76.7 $\pm$ 0.4     | 70.8 $\pm$ 0.6     | 76.5 $\pm$ 0.5     |
> | SGC             | 81.0 $\pm$ 0.2     | 71.3 $\pm$ 0.5     | 78.9 $\pm$ 0.5     |
> | GAT             | 83.0 $\pm$ 0.7     | 72.5 $\pm$ 0.6     | 79.0 $\pm$ 0.3     |
> | PPRGo           | 82.4 $\pm$ 0.3     | 71.3 $\pm$ 0.5     | 80.0 $\pm$ 0.4     |
> | APPNP           | 83.3 $\pm$ 0.5     | 71.8 $\pm$ 0.5     | 79.7 $\pm$ 0.3     |
> | DAGNN           | **84.4 $\pm$ 0.6** | 73.3 $\pm$ 0.6     | **80.5 $\pm$ 0.5** |
> | **NAFS-mean**   | 84.2 $\pm$ 0.6     | 73.2 $\pm$ 0.5     | **80.5 $\pm$ 0.5** |
> | **NAFS-max**    | 83.8 $\pm$ 0.7     | 73.4 $\pm$ 0.6     | 80.4 $\pm$ 0.6     |
> | **NAFS-concat** | 84.1 $\pm$ 0.6     | **73.5 $\pm$ 0.4** | 80.3 $\pm$ 0.4     |
>
>
> The table shows that our method achieves comparable predictive accuracy as one of the state-of-the-art methods DAGNN. However, since the node embeddings are generated before training, our method avoids performing recursive feature smoothing at each training epoch and storing the entire adjacency matrix on GPU. In this way, our method is more scalable and efficient to apply on large graphs than most state-of-the-art methods like DAGNN.
>
> As suggested by the reviewer, we have added the performance comparison on node classification task in Table 4 of the revised manuscript.
>
> [1] Kipf, Thomas N., and Max Welling. [*Variational graph auto-encoders.*](https://arxiv.org/pdf/1611.07308) NIPS, 2016.
>
> [2] Wang, Chun, et al. [*Attributed Graph Clustering: a Deep Attentional Embedding approach.*](https://www.ijcai.org/proceedings/2019/0509.pdf) IJCAI, 2019.
>
> [3] Cui, Ganqu, et al. [*Adaptive graph encoder for attributed graph embedding.*](https://dl.acm.org/doi/pdf/10.1145/3394486.3403140) SIGKDD, 2020.

---

> > ### Author Response · Authors · 2021-11-18
> > **Response to Reviewer 1, Part 2**
> >
> > ### Q3: How the weights help avoid converging.
> >
> > As suggested by the reviewer, we explain the motivation of the weighting scheme by formally demonstrating how it impacts final representation when $\small K$ grows from a more theoretical perspective. Recall the final representation $\small \hat{\mathbf{X}} = \sum\limits\_{k=0}\limits^{K}\mathbf{W}(k)\hat{\mathbf{A}}^{k}\mathbf{X}$, which combines the representations obtained at each smoothing step. Here, $\small \mathbf{W}(k)$ denotes the importance of the $k$-th step representation in constituting the final representation and there is a constraint that $\small \sum\limits\_{k=0}\limits^{K}\mathbf{W}(k)=1$.
> >
> > We examine the distance from the final representation (when K  approaches infinity) $\hat{\mathbf{X}}\_i$ of node $i$ to its over-smoothed representation $\small [\hat{\mathbf{A}}^{\infty}\mathbf{X}]\_i$:
> > \begin{equation}
> > \begin{split}
> > \small
> > \||\hat{\mathbf{X}}\_i-[\hat{\mathbf{A}}^{\infty}\mathbf{X}]\_i\||\_{2}
> > \&= \small  \lim\_{K\to \infty}\||\sum\_{k=0}^{K}\omega\_i(k)*([\hat{\mathbf{A}}^{k}X]\_{i}-[\hat{\mathbf{A}}^{\infty}X]\_{i})\||\_{2}\\
> > \& \small \le \lim\_{K\to \infty} \sum\_{k=0}^{K}\omega\_i(k)*\||[\hat{\mathbf{A}}^{k}X]\_{i}-[\hat{\mathbf{A}}^{\infty}X]\_{i}\||\_{2}=\lim\_{K\to \infty}\sum\_{k=0}^{K}\omega\_i(k)*D\_i(k).
> > \end{split}
> > \end{equation}
> >
> > To better understand of the effects that different weighting schemes impose, we first simply set the weight uniformly as $\small \mathbf{W}(k)=\frac{1}{K+1}$. By combining Theorem A.1 in Appendix A, we have:
> > \begin{equation}
> > \begin{split}
> > \small
> > \||\hat{\mathbf{X}}\_i-[\hat{\mathbf{A}}^{\infty}\mathbf{X}]\_i\||\_{2} \le \lim\_{K\to \infty} \sum\_{k=0}^{K}\omega\_i(k)*D\_i(k)
> > \&= \small \lim\_{K\to \infty} \frac{1}{K+1}\sum\_{k=0}^{K}D\_i(k)\\
> > \& \small \le \lim\_{K\to \infty} \frac{1}{K+1}\sum\_{k=0}^{K}\lambda\_{2}^{k}\sqrt{\frac{cdx}{\tilde{d}\_i}}\\
> > \& \small  \le \lim\_{K\to \infty} \frac{1}{K+1}\frac{1}{1-\lambda\_{2}}\sqrt{\frac{cdx}{\tilde{d}\_i}}=0,
> > \end{split}
> > \end{equation}
> > where $\small cdx=\sum\limits\_{j=1}\limits^{n}(\tilde{d\_{j}}\||x\_{j}\||\_{2}^{2})$ is a constant independent of $\small k$, and $\small 0<\lambda\_{2}<1$ denotes the second largest eigenvalue of the normalized adjacency matrix. We see that the final representation
> > will be over-smoothed if only adopting uniform weights that $\small \mathbf{W}(k)=\frac{1}{K+1}$.
> >
> > We next show our distance-aware weighting scheme actually prevents the final node representations from converging to the identical one.
> > The basic idea of our weighting scheme is assigning weights *proportional to the over-smoothing distance $\small D\_{i}(k)$* of the features at each smoothing step, and features far from the stationary state should contribute more to the final node representations. Let the weights be the $l\_1$ normalization of the over-smoothing distance of features at different smoothing steps, i.e., $\small \omega\_i(k) = D\_i(k)/(\sum\_{l=0}^{K}D\_i(l))$.
> > Let $\small X\_i^k = [\hat{\mathbf{A}}^{k}X]\_{i}-[\hat{\mathbf{A}}^{\infty}X]\_{i}$, we have:
> > \begin{equation}
> > \begin{split}
> > \small
> > \||\hat{\mathbf{X}}\_i-[\hat{\mathbf{A}}^{\infty}\mathbf{X}]\_i\||\_{2}
> > \&=\small \lim\_{K\to \infty}\omega\_i(0)\||X\_i^0+\sum\_{k=1}^{K}\frac{\omega\_i(k)}{\omega\_i(0)}X\_i^k\||\_{2}.
> > \end{split}
> > \end{equation}
> >
> > In real-world graphs, nodes have different initial features. There is little chance that the combination of a node's neighboring features both lies in the opposite direction and is of the same norm of the node's initial feature. Suppose that there exists a constant $\small \epsilon > 0$ satisfying $\small \min\left(\{D\_i(0),\||X\_i^0+\sum\_{k=1}^{K}\frac{\omega\_i(k)}{\omega\_i(0)}X\_i^k\||\_{2}\}\right) \geq \epsilon$ for node $\small i$, we have:
> > \begin{equation}
> > \begin{split}
> > \small
> > \||\hat{\mathbf{X}}\_i-[\hat{\mathbf{A}}^{\infty}\mathbf{X}]\_i\||\_{2} \& \geq  \frac{\epsilon^2}{\lim\_{K\to \infty}\sum\_{k=0}^{K}D\_i(k)} \ge \frac{\epsilon^2}{\frac{1}{1-\lambda\_{2}}\sqrt{\frac{cdx}{\tilde{d}\_i}}} > 0.
> > \end{split}
> > \end{equation}

---

> > > ### Author Response · Authors · 2021-11-18
> > > **Response to Reviewer 1, Part 3**
> > >
> > > Thus, when $\small K$ goes to infinity, we can prevent the node representations from reaching the stationary state (the distance bound depends on the node degree $d_i$ and the initial feature $\small \mathbf{X}_i$). Note that the representation of $k$-th smoothing step $\small \mathbf{X}^{(k)}$ practically achieves the stationary state much earlier than the infinity-th smoothing step we use in the theoretical analysis. Therefore, we use the softmax normalization in Equation 5 of Section 4.1 to produce a slightly larger bias towards features with longer distances to the stationary state.
> > > We also provide the node clustering performance of the $\small l_1$ normalization in the table below, which is also a specific instance of our distance-aware weighting scheme. The following tables evaluated on the Cora dataset show that the softmax normalization usually performs better than the $\small l_1$ normalization.
> > >
> > > The results of NAFS-mean:
> > >
> > > | Normalization     | ACC  | NMI  | ARI  |
> > > | ----------------- | ---- | ---- | ---- |
> > > | $\small l_1$ norm | 68.2 | 54.5 | 47.4 |
> > > | softmax           | 70.4 | 56.6 | 48.0 |
> > >
> > > The results of NAFS-max:
> > >
> > > | Normalization     | ACC  | NMI  | ARI  |
> > > | ----------------- | ---- | ---- | ---- |
> > > | $\small l_1$ norm | 69.8 | 56.5 | 47.0 |
> > > | softmax           | 70.8 | 56.6 | 49.0 |
> > >
> > > The results of NAFS-concat:
> > >
> > > | Normalization     | ACC  | NMI  | ARI  |
> > > | ----------------- | ---- | ---- | ---- |
> > > | $\small l_1$ norm | 67.3 | 52.7 | 43.9 |
> > > | softmax           | 75.4 | 58.6 | 53.8 |
> > >
> > > ### Q4: Minor corrections
> > > Thank you for reminding us of these typos. We have corrected them in the revised manuscript.

---

> ### Author Response · Authors · 2021-11-22
> **Response to Reviewer 1**
>
> We really appreciate your helpful and valuable reviews, especially the advice of:
> 1) Adding the discussion with NDLS.
> 2) Adding the evaluation in the node classification task.
> 3) Explaining how the weighting scheme impacts final representations from a theoretical perspective.
>
> We have carefully responded to your questions and added the experiments as suggested.
>
> We hope to address all your points, and we are happy to respond if you have additional comments or concerns on our responses.
>
> Respectfully,
>
> Paper608 Authors

---

> ### Author Response · Authors · 2021-11-23
> **Answer for theoretical concerns to Reviewer 1**
>
> To address the problem of "lack of a theoretical interpretation for the weighting scheme" as suggested by Reviewer 1, we have improved the theoretical and methodological significance of this work in Appendix. A of our revised manuscript.
>
> Concretely, we propose a new methodological principle to combine representations over different smoothing steps: Assigning weights proportional to the over-smoothing distance of the features at each smoothing step $k$, i.e., $\omega_i(k) \propto D_i(k)$. It is worth pointing out that the speed to approach the over-smoothing stationarity is *heterogeneous* across nodes (as implied by our theorem A.1 in Appendix. A). Therefore, the methodological contribution lies in a new **over-smoothing-distance-aware** and **node-adaptive** weight assignment scheme, which is fundamentally different from the traditional routine of the smooth-blind and fixed scheme.
>
> A key theoretical contribution is that the proposed methodological principle is **theoretically proved** to prevent
> over-smoothing as the smoothing step goes to infinity. By contrast, the final representation will be over-smoothed if only adopting fixed and uniform weights. The Appendix. A in the revised manuscript provides the detailed theoretical analysis.
>
> The detailed theoretical interpretation can be found in our earlier response. Here, we provide a **theoretical analysis sketch** to understand how our weighting principle actually impacts final representations from a theoretical perspective.
> We first give an upper bound to the over-smoothing distance at each step $k$, showing $D_{i}(k) \to 0$ quickly as $k$ increases.
> Next, we show simply setting the weight uniformly as $\omega_i(k) = 1/(K+1)$ would over-smooth the final representation $\hat{\mathbf{X}}_i$ of a node $i$ ( i.e., $||\hat{\mathbf{X}}\_i-[\hat{\mathbf{A}}^{\infty}\mathbf{X}]\_i||\_{2} \to 0$), because the uniform weight would make over-smoothed features dominate the final representation. By contrast, we show that distance-aware weighting scheme actually prevents the final node representations from converging to the identical one (i.e., there is a constant $\epsilon > 0$ making
> $||\hat{\mathbf{X}}\_i-[\hat{\mathbf{A}}^{\infty}\mathbf{X}]\_i||\_{2} > \epsilon$). This is because the weight decreases at the same rate with over-smoothing distance of smoothed features, excluding over-smoothed features from the final representation while retaining all feature that are not over-smoothed (i.e., $\omega\_i(k)>0$ given $D\_i(k) > 0$).
>
> The empirical results in the original manuscript further demonstrate the validity of our principle as it could fully leverage features at a different level of localities in a node-adaptive manner.
>
> We hope this answer could address your concern about the theoretical interpretation, and we are happy to respond if you have additional comments or concerns on our responses.
>
> Respectfully,
>
> Paper608 Authors

---

### Decision · Program_Chairs · 2022-01-20

**Decision:**

Reject

**Comment:**

The paper proposes NAFS (Node-Adaptive Feature Smoothing), which constructs node representations by only using smoothing without parameter learning.  The authors first provide a formulation for the smoothing operator. They then define over-smoothing distance to assess how much a node is close to the stationary state. Finally, they use the over-smoothing distance to calculate a smoothing weight for each node. Experiments are conducted to verify the efficacy.

Strength
* The paper tackles the problem of over-smoothing, which is a well-known issue in GNN.
* The solution appears to be effective.
* The paper is generally clearly written.

Weakness
* The novelty and significance of the work might not be enough. Aspects of the contributions exist in prior work.

---

Additional experiments have been conducted during the rebuttal. The reviewers appreciate the efforts.

After rebuttal：

Reviewer SHxg increased the score accordingly.

Reviewer w2Qg says “Given the concerns proposed by the other reviewers, I adjusted my score.”

Reviewer YM4P says “I read the rebuttal and slightly increased my score.”